# Atomic structures of Coxsackievirus A6 and its complex with a neutralizing antibody

Longfa Xu[1], Qingbing Zheng[1], Shaowei Li[1], Maozhou He[1], Yangtao Wu[1], Yongchao Li[1], Rui Zhu[1], Hai Yu[1], Qiyang Hong[1], Jie Jiang[1], Zizhen Li[1], Shuxuan Li[1], Huan Zhao[1], Lisheng Yang[2], Wangheng Hou[1], Wei Wang[1], Xiangzhong Ye[2], Jun Zhang[1], Timothy S. Baker[3], Tong Cheng[1], Z. Hong Zhou[4,5], Xiaodong Yan[1,3] & Ningshao Xia[1]

Coxsackievirus A6 (CVA6) has recently emerged as a major cause of hand, foot and mouth disease in children worldwide but no vaccine is available against CVA6 infections. Here, we demonstrate the isolation of two forms of stable CVA6 particles-procapsid and A-particle-with excellent biochemical stability and natural antigenicity to serve as vaccine candidates. Despite the presence (in A-particle) or absence (in procapsid) of capsid-RNA interactions, the two CVA6 particles have essentially identical atomic capsid structures resembling the uncoating intermediates of other enteroviruses. Our near-atomic resolution structure of CVA6 A-particle complexed with a neutralizing antibody maps an immune-dominant neutralizing epitope to the surface loops of VP1. The structure-guided cell-based inhibition studies further demonstrate that these loops could serve as excellent targets for designing anti-CVA6 vaccines.

[1] State Key Laboratory of Molecular Vaccinology and Molecular Diagnostics, National Institute of Diagnostics and Vaccine Development in Infectious Diseases, School of Public Health, Xiamen University, Xiamen 361102, PR China. [2] Department of Research & Development Beijing Wantai Biological Pharmacy Enterprise Co., Ltd., Beijing 102206, PR China. [3] Department of Chemistry and Biochemistry and Division of Biological Sciences, University of California-San Diego, San Diego, CA 92093-0378, USA. [4] The California NanoSystems Institute (CNSI), UCLA, Los Angeles, California 90095, USA. [5] Department of Microbiology, Immunology and Molecular Genetics, UCLA, Los Angeles, California 90095, USA. Longfa Xu, Qingbing Zheng and Shaowei Li contributed equally to this work. Z. Hong Zhou and Timothy S. Baker jointly supervised this work. Correspondence and requests for materials should be addressed to T.C. (email: tcheng@xmu.edu.cn) or to X.Y. (email: xdy@ucsd.edu) or to N.X. (email: nsxia@xmu.edu.cn)

Coxsackievirus A6 (CVA6), a member of the human Enterovirus species A (family *Picornaviridae*, genus *Enterovirus*), has recently emerged as one of the major pathogens of hand, foot, and mouth disease (HFMD) in children worldwide[1]. CVA6 was associated with many HFMD outbreaks in Asia[2], Europe[3], and North America[4, 5]. Unlike the 'classical' enteroviruses, including enterovirus 71 (EV71) and Coxsackivirus A16 (CVA16), CVA6 infection can lead to many atypical clinical symptoms such as vesiculobullous eruption[6], onychomadesis[7] or herpangina[8]. In addition, CVA6 can cause serious central nervous system disorders such as aseptic meningitis and brainstem encephalitis[9, 10]. Indeed, CVA6 is considered to be as pathogenic as, if not more than, EV71 and CVA16. Although highly effective vaccines exist for EV71[11], no vaccine is available for the global public health threats imposed by CVA6, which underscores the critical need for developing an effective vaccine.

Atomic structures of EV71 and CVA16 have helped propel progress in vaccine design targeted against these picornavirus infections[12, 13]. The structures of different types of EV71, CVA16 and other picornavirus particles and their immune-complexes have been extensively studied at or near atomic resolution by means of cryo-electron microscopy (cryoEM) or X-ray crystallography[12, 14–19]. Such structural knowledge has also provided key insights into the picornaviral life cycle, including mechanisms of receptor binding and accompanying conformational changes, cell entry and genome release, and viral assembly. Two types of stable particles are typically isolated: mature virions (containing RNA genome) and procapsid (empty, lacking RNA genome)[12, 20]. The mature particle of typical picornaviruses contains sixty copies each of four unique structural proteins (VP1, VP2, VP3, and VP4) arranged with pseudo $T = 3$ symmetry[21, 22]. VP1, VP2, and VP3 all have a compact core structure consisting of a classical, β-sandwich, jelly-roll fold, which forms the bulk of the capsid and each of these protein subunits also have several loops that decorate the outer surface of the capsid. Unlike the other viral proteins, VP4 adopts an extended conformation on the inner surface of the mature capsid[21, 22]. Upon binding of cellular receptor(s) at the surface depression (canyon) encircling each icosahedral 5-fold vertex[12, 23], mature virus releases the lipid moiety (pocket factor) buried in a hydrophobic pocket formed by VP1 beneath the canyon. Loss of pocket factor allows the pocket to collapse and a channel close to the icosahedral 2-fold axis to open, leading to an expanded, third type of particle, called uncoating intermediate or A-particle[14, 15, 24–26]. Although A-particles are characterized by their inherent instability, the structures of some of them have been successfully studied by examining particles to which receptors were bound[26], or by treating them with heat or low pH[15, 27, 28]. The structures of these A-particles show several distinctive features including expanded capsids, open channels at the 2-fold axes, and capsid-RNA interactions[15, 26–28].

Though empty and full CVA6 particles have been isolated, there has not been systematic characterization for any of these

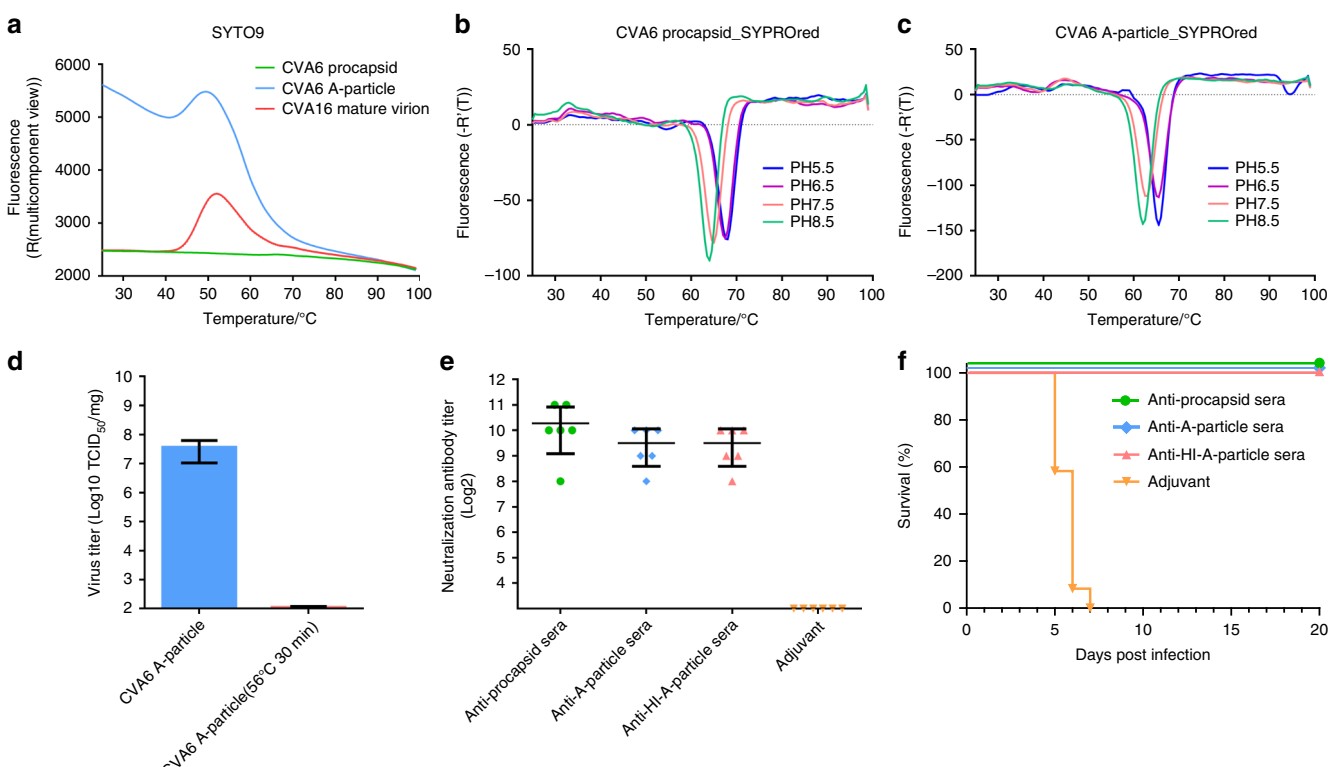

**Fig. 1** Stability infectivity and immunogenicity of CVA6 particles. **a** PaSTRy assays of CVA6 procapsids, A-particles and CVA16 mature virions. The high fluorescence level throughout the whole temperature range indicates the genome in CVA6 A-particle is more exposed than the one in CVA16 mature virion. In addition, the genomes in CVA6 and CVA16 virions both reach their maximum exposures at ~52 °C, presumably due to capsid expansion triggered by heat treatment. **b, c** The thermal stability of CVA6 procapsid **b** and A-particle **c** measured at pH values ranging from 5.5 to 8.5. The first derivatives (-R' (T)) vs temperature are plotted at different pHs. Both particles are most stable at pH 5.5. **d** The infectivity of the CVA6 A-particle ($3.6 \times 10^7$ TCID$_{50}$/mg) estimated by TCID$_{50}$ assays on RD cells (mean ± s.d.). **e** In vitro neutralizing titer of antisera of CVA6 particles. Four groups of mice, each including six female BALB/c mice, were vaccinated subcutaneously with CVA6 procapsid, A-particle, heating-inactivated (HI) A-particle and adjuvant (control) at days 0 and 21. Blood samples of the mice were assayed at day 35 and the level of neutralization antibody titer of all mice in each group were plotted (mean ± s.d.). **f** In vivo protective efficacy of antisera of CVA6 particles. The infected mice were treated with antisera against procapsid, A-particle and HI-A-particle respectively and all three experimental groups had a 100% survival rate compared to 0% for the adjuvant control group

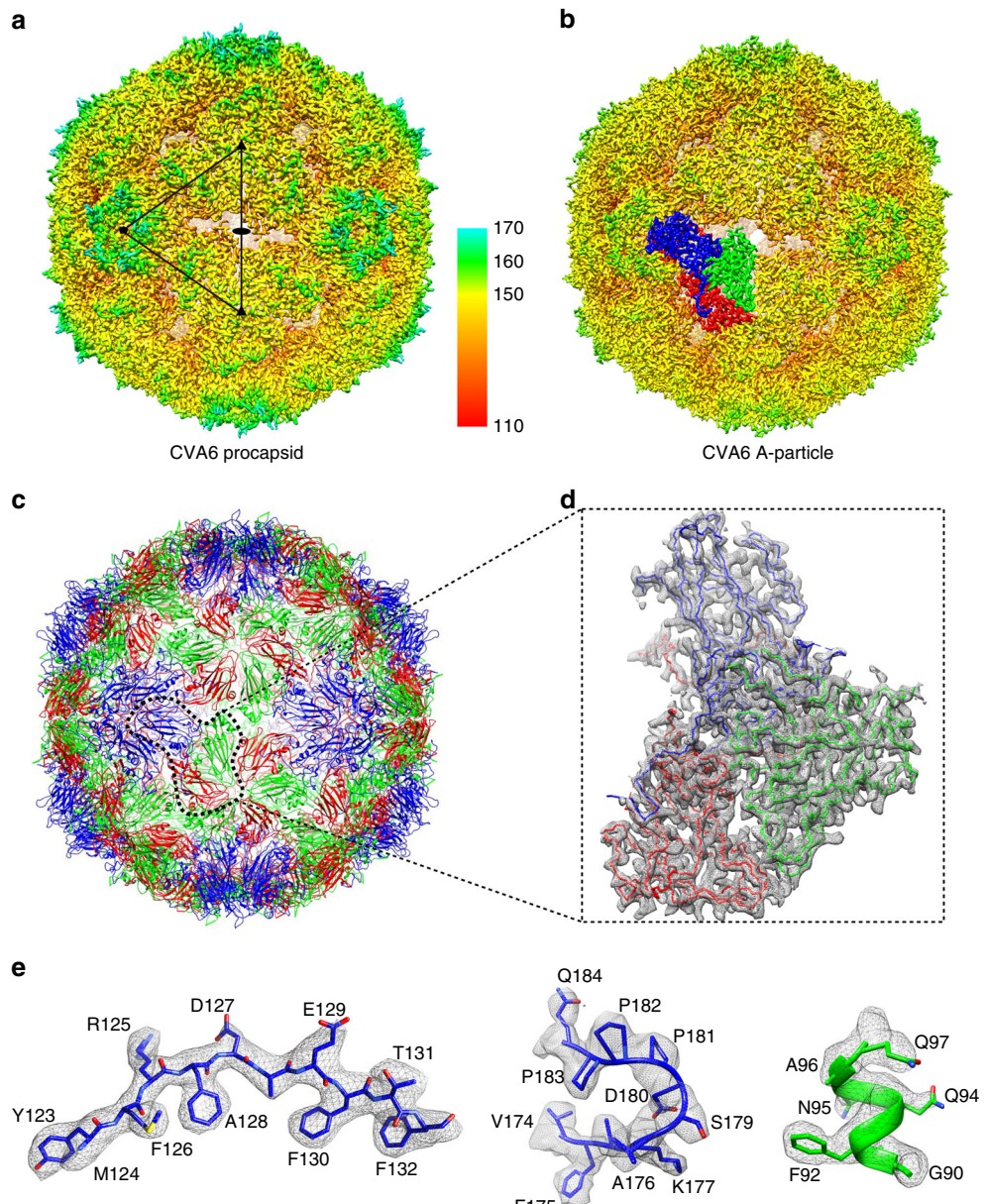

**Fig. 2** CryoEM structures of CVA6 procapsid and A-particle. **a**, **b** Iso-contoured views (radially colored) of cryoEM density maps of procapsid **a** and A-particle **b** viewed along a 2-fold axis. One icosahedral asymmetric unit is marked with a *black triangle* in **a** and a single icosahedral protomer is drawn in **b**, **c**. The maps of the procapsid and A-particle are essentially identical. **c** Atomic model of the A-particle capsid. The ribbon diagram demonstrates the pseudo *T* = 3 arrangement of component proteins: VP1 (*blue*), VP2 (*green*) and VP3 (*red*). This color scheme is used throughout the manuscript unless noted otherwise. **d** One segmented asymmetric unit of the A-particle density map (*gray*) fitted with its atomic model (Cα backbones) to highlight the densities attributed to each individual protein. **e** The quality of the A-particle density map (*gray*) is illustrated by the fit of backbone and side chains for three separate structural motifs: a β-strand (VP1), a loop (VP1) and an α-helix (VP2)

particles to date and whether they are similar to or different from the above three types of picornavirus particles remains unclear[29]. Although CVA6 shares high (~67%) amino-acid identity with EV71 and CVA16[30], CVA6 recognizes a different cellular receptor[31, 32], suggesting CVA6 may differ in its mode of infection, or perhaps even in its local structures involved in infection. Here, we demonstrate the isolation of two stable CVA6 particles with excellent natural antigenicity: one empty (CVA6 procapsid) and the other infectious (CVA6 A-particle). The capsid structures of CVA6 procapsid and A-particle at near atomic (~3 Å) resolution both closely resemble those of the uncoating intermediates of other enteroviruses. In addition, structural and functional studies of the CVA6 A-particle complexed with the Fab of a

neutralizing antibody identify four surface loops on VP1 that can be targeted for vaccine design.

## Results

**CVA6 procapsids and A-particles as viable vaccine candidates.** CVA6 was grown in human rhabdomyosarcoma (RD) cells and purified following routine protocols employing centrifugation, ultracentrifugation, and ultrafiltration. Two types of particles were purified for subsequent biochemical and structural analyses (Supplementary Fig. 1a). The top band exhibited a λ260/λ280 absorbance ratio of 0.71, and the particles in that band sedimented with a coefficient of 84S (Supplementary Fig. 1b), had a

**Table 1 CryoEM data collection and atomic models refinement statistics**

| | CVA6 procapsid | CVA6 A-particle | CVA6 A-particle-1D5 complex |
|---|---|---|---|
| *Data collection* | | | |
| EM equipment | Tecnai F30 | Tecnai F30 | Tecnai F30 |
| Voltage (kV) | 300 | 300 | 300 |
| Detector | Falcon II | Falcon II | Falcon II |
| Pixel size (Å) | 1.128 | 1.128 | 1.128 |
| Electron dose (e$^-$/Å$^2$) | 25 | 25 | 25 |
| Defocus range (μm) | 1.3–4.0 | 1.0–4.3 | 0.7–4.1 |
| *Reconstructure* | | | |
| Software | Relion 1.4 | Relion 1.4 | Relion 1.4 |
| Number of used particles | 10,749 | 7,152 | 12,067 |
| Final resolution (Å) | 3.3 | 3.1 | 3.8 |
| *Model building* | | | |
| Software | Coot | Coot | Coot |
| *Refinement* | | | |
| Software | Phenix | Phenix | Phenix |
| *Model statistics* | | | |
| Correlation coefficient (around atoms) | 0.899 | 0.831 | 0.926 |
| No. of atoms protein | 4,799 | 4,909 | 6,733 |
| Average B factor (Å$^2$) | −218.67 | −163.92 | −223.15 |
| *R.m.s.d.* | | | |
| Bond lengths (Å) | 0.008 | 0.009 | 0.011 |
| Bond angles (°) | 1.19 | 1.27 | 1.21 |
| *Ramachandran plot* | | | |
| Favored (%) | 97.13 | 96.06 | 94.07 |
| Allowed (%) | 2.87 | 3.94 | 5.57 |
| Outliers (%) | 0.00 | 0.00 | 0.36 |
| Rotamer outliers (%) | 0.00 | 0.00 | 0.00 |

A-particles remain stable throughout this pH range and are most stable (melting point of ~65 °C) at pH 5.5 (Fig. 1b, c), which is comparable to the acidic environment in the human gut during infection. In addition, the high infectivity of the CVA6 A-particle was measured to be $3.6 \times 10^7$ TCID$_{50}$/mg, which demonstrates that A-particles can infect cells efficiently (Fig. 1d). The above results indicate that the isolated CVA6 A-particle is biochemically similar to the uncoating intermediates of other picornaviruses, but functions as a mature virus that has a stable structure and is highly infectious.

Potent immunogenicity is an additional prerequisite for a good vaccine candidate. To analyze the immunogenicity of CVA6 virus, mice were vaccinated with purified procapsids, A-particles or heat-inactivated (HI) A-particles. Antisera elicited from all three groups showed high and comparable neutralizing titers against CVA6, indicating that all three of these particles exhibit similar structures and antigenicity (Fig. 1e). In addition, the therapeutic efficacy of antisera in vivo against CVA6 infection was assessed using 1-day-old mice model. Three different antisera all showed 100% treatment potential ($P < 0.001$) against a dose (75 LD$_{50}$) of CVA6 infection, whereas the control group started to show signs of illness 5 d.p.i. and all died within 7 d.p.i. (Fig. 1f). Hence, the similar antigenicity and potent therapeutic efficacy of their antisera qualify both procapsid and A-particles as excellent vaccine candidates against CVA6 infection.

**Atomic models of CVA6 procapsid and A-particle**. Purified procapsid and A-particle samples were vitrified and imaged by low-dose technique (Supplementary Fig. 2a, b). A total of 10,749 procapsid and 7152 A-particle images were extracted and subjected to three-dimensional (3D) reconstruction, respectively, yielding final density maps at estimated resolutions of 3.3 and 3.1 Å (Fig. 2a, b, Supplementary Figs 3a–f and 5, Table 1 and Supplementary Movie 1). The density maps of the procapsid and A-particle reveal essentially identical (0.96 correlation coefficient), pseudo $T = 3$ icosahedral capsid structures, which explains why the two particles share similar antigenicity. Their sizes are the same too, with diameters being 290, 310, 330 Å, along their 2-, 3-, and 5-fold axes, respectively (Supplementary Fig. 4). Densities attributable to residue backbones and side chains, especially bulky ones, were recognizable in both maps (Fig. 2e and Supplementary Movie 1). The obvious difference between two density maps is the density of genomic RNA, which is present in A-particles but absent in procapsids (Supplementary Fig. 4). Atomic models of their biological protomers (VP0 + VP1 + VP3 for the procapsid and VP1 + VP2 + VP3 for the A-particle) were built manually and subsequently refined by Phenix (Supplementary Fig. 5 and Table 1). The structural similarity between the two atomic models was deemed quite high as judged by their superposition (Supplementary Fig. 6c) and confirmed by an r.m.s.d. value of 0.4 Å between the two models. Though the procapsid was shown by SDS–PAGE to contain VP0 (containing the sequences of VP2 and VP4), only the VP0 sequence corresponding to VP2 is well resolved in the density map and hence successfully modeled (Fig. 2d and Supplementary Movie 1), indicating the one corresponding to VP4 is flexible as that in previously reported EV71 procapsid[12]. Aside from this flexible region, both models contain several additional coincidental flexible segments (Supplementary Fig. 6 and Supplementary Table 1). Among these segments, seven are identical and the others vary from 2 to 10 amino acid residues (Supplementary Fig. 6a, b and Supplementary Table 1). The first 70 residues at the N-terminus of VP1 are missing in both models; The AB loop of VP2 in the A-particle contains a flexible segment of 10 a.a., but 20 a.a. in the procapsid; In VP3, one small segment (a.a.74–77) is missing in the procapsid, whereas the

capsid protein composition consistent with the presence of VP0, VP1, and VP3 in sodium dodecyl supfate–polyacrylamide gel electrophoresis (SDS–PAGE) gel electrophoresis (Supplementary Fig. 1d, lane 2), and showed an absence of packaged genome as verified by negative stain electron microscopy (Supplementary Fig. 1e). These characteristics are all consistent with this band being composed of purified procasids. The bottom band had a λ260/λ280 absorbance ratio of 1.65 with particles sedimenting at 128S (Supplementary Fig. 1c) and composed of capsid proteins VP1, VP2, and VP3 but no VP4 (Supplementary Fig. 1d, lane 3), and contained packaged genome as verified by negative staining (Supplementary Fig. 1f). It is well known that procapsids and mature virions are typically the dominant types of picornavirus particles harvested in vivo[12]. Mature virus (160S) contains one copy of the genomic RNA and 60 copies each of capsid proteins VP1, VP2, VP3, and VP4. Hence, the bottom band unexpectedly contained particles more like uncoating intermediates (or A-particles) as opposed to mature virions of other picornaviruses. The thermal stability tests also confirmed this observation (Fig. 1a). Of note, repeated rounds of isolation and purification produced the same two bands of distinct but stable particles, and as such we concluded that the CVA6 A-particles in the bottom band represent a picornavirus anomaly.

The biological stability and infectivity of CVA6 procapsids and A-particles were tested to determine if either or both of these particles could serve as vaccine candidates. The protein melting temperatures of both particles gradually rise as the pH is progressively lowered from 8.5 to 5.5 (Fig. 1b, c). Procapsids and

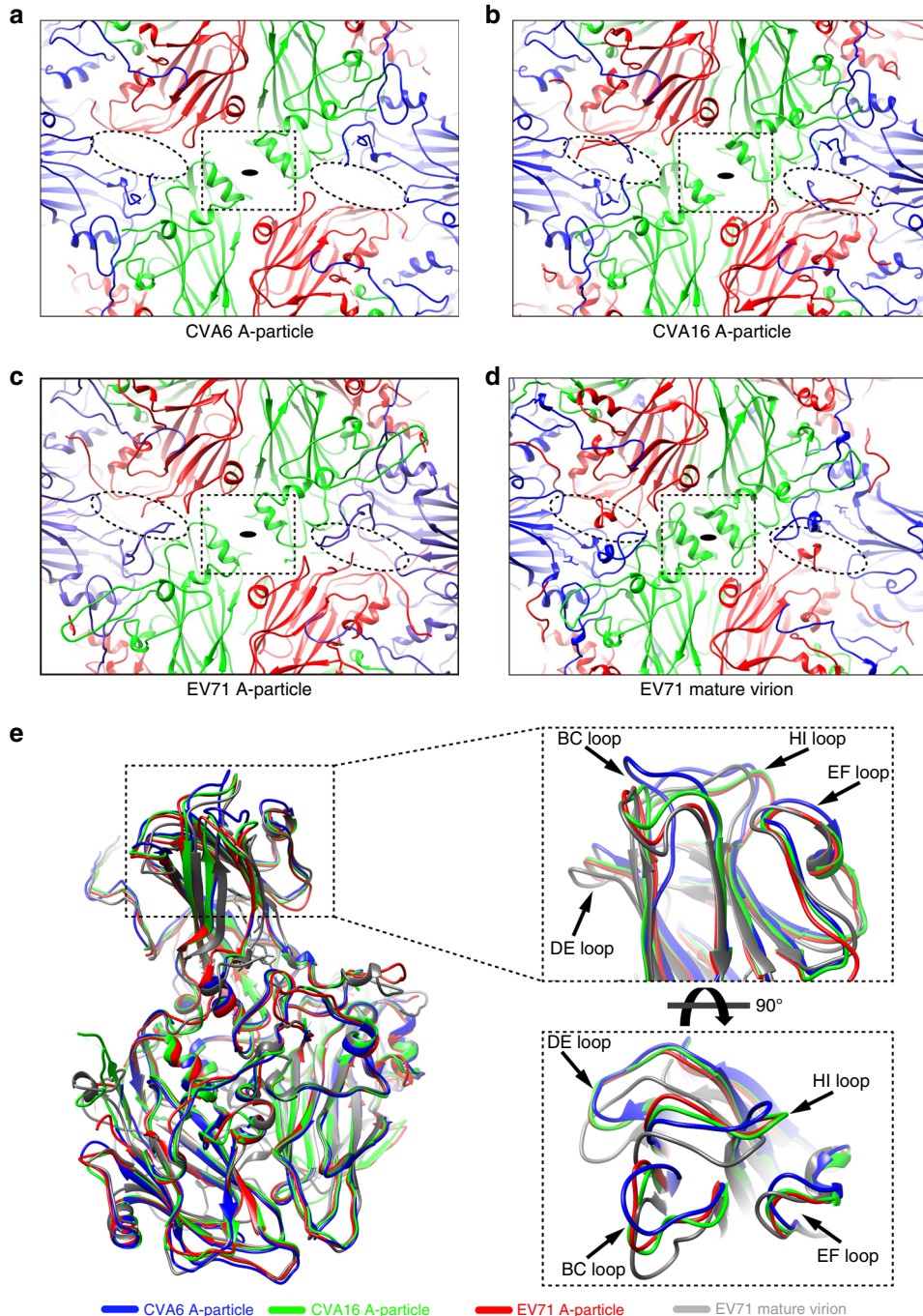

**Fig. 3** Structural comparisons of CVA6, CVA16, and EV71. **a–d** Close up views centered at a 2-fold axis (*black ellipses*) of atomic models of CVA6 A-particle **a**, CVA16 A-particle **b**, EV71 A-particle **c**, and EV71 virion **d**. The 2f-channels (*dashed squares*) appear wide **a**, **b**, narrow **c**, and closed **d**. These comparisons also demonstrate that the q3f-channels (*dashed ellipses*) are un-obstructed **a**, obstructed **b**, **c**, and closed **d**. **e** Superposition of VP1 surface loops of CVA6 A-particle, CVA16 A-particle, EV71 A-particle and EV71 mature virion. All four of these loops, especially the BC loop, exhibit prominent differences among the different particles

corresponding segment in the A-particle is intact (Supplementary Fig. 6 and Supplementary Table 1). Atomic models of entire capsids (Fig. 2c), except for the disordered regions identified above, were generated using the program Chimera[33].

The atomic model of the CVA6 A-particle reveals that it has a similar capsid structure compared to other known picornavirus A-particles (Fig. 3 and see also structure-based sequence alignments in Supplementary Fig. 7). Indeed, the Cα positions of 621 aligned residues of the CVA16 A-particle (PDB code 4JGY)[14] and 618 aligned residues of the EV71 A-particle (PDB

code 4N43)[25] when compared to the CVA6 A-particle gave r.m.s.d. values of 0.8 and 0.9 Å, respectively (Supplementary Table 2). The CVA6 A-particle contains characteristic structural features that are typically attributed solely to picornavirus A-particles. These include large, open channels at the icosahedral 2-fold (2f-channel) and quasi 3-fold axes (q3f-channel) (Fig. 3a), and a collapsed hydrophobic pocket beneath VP1 'canyon'. However, close inspection of the CVA6 model also reveals several significant structural differences from CVA16 and EV71 A-particles as well as EV71 virions (PDB code 3VBS) (Fig. 3b, d).

The 2f-channel in CVA6 has dimensions of 36 Å × 8 Å, which are both similar to that in the CVA16 A-particle and the EV71 A-particle (Fig. 3b, c), whereas the channel is closed in the EV71 virion (Fig. 3d). The q3f-channel in CVA6 is 23 Å × 13 Å in size, but it is more obstructed in CVA16 and EV71 A-particles and in EV71 virion. All three capsid proteins in the CVA6 A-particle show regions of poor densities near the q3f-channel, such as the GH loop (a.a. 205–217) in VP1, the AB (a.a. 43–52) and EF (a.a. 138–145) loops in VP2, and the GH loop (a.a. 173–188) in VP3 (Supplementary Fig. 6 and Supplementary Table 1).

The most significant differences among the three picorna-viruses used for comparison occur at the four surface loops (BC, DE, EF, and HI) of VP1 (Fig. 3e and Supplementary Movie 1). It is well documented that five copies of VP1 associate together to form a prominent protrusion on the capsid surface 'cooling tower'[34] at each 5-fold vertex, and the four loops together form the topmost portion of the 5-fold channel. The r.m.s.d. values in the Cα positions for matched residues in these loops are 1.8, 1.9, and 2.8 Å for comparison of the CVA16, EV71 A-particles and EV71 mature virus, respectively, while the value is just 0.3 Å in comparison to CVA6 procapsid (Supplementary Table 2); in particular, the BC loop exhibits the largest differences among these four particles, and the r.m.s.d. in the Cα positions for matched residues in BC loop is 2.3, 2.0 and 3.2 Å, respectively (Supplementary Table 2).

**Capsid-RNA interactions involving the N-terminus of VP3.** In contrast to other known enteroviruses, CVA6 A-particles display unique interactions between its capsid and the packaged genome[14, 15, 26]. The cryoEM density map and atomic model of the A-particle together reveal that the N-termini of VP2 and VP3 interact with the genomic RNA near the icosahedral 2- and 5-fold axes, respectively (Fig. 4a, b). Notably, the first 29 residues of the VP2 N-terminus are not included in the atomic model since these residues in each of the 60 copies of VP2 in the symmetrized density map are not likely to interact equivalently with the single copy of the genome. Two residues in VP2 lie closest to the genome: E37 and W38 (Fig. 4c, d). The fact that VP2 W38 residues interact with the RNA near the 2-fold axes has also been noted in other picornavirus A-particles[26–28]. In addition, strong interactions between the N-termini of VP3 and the genomic RNA were clearly apparent in the CVA6 A-particle density map. The N-termini of five VP3 subunits extend beneath the VP1 'cooling tower' to form a well-defined annulus with the key residue P3 making close interactions with the genomic RNA (Fig. 4e). Similar interactions between the VP3 N-termini and the genome at the icosahedral 5-fold axes have been reported in picornavirus Ljungan virus, HPeV1, HPeV3[35] and human rhinovirus[28] but have never been observed in any of the enterovirus species. However, though VP1-mediated RNA contacts have been reported in other picornaviruses[14, 15, 25], none were detected in the CVA6 A-particle in this study.

**Atomic model of CVA6 A-particle-1D5 immune-complex.** We also used cryoEM and 3D image reconstruction methods to investigate the structure of the CVA6 A-particle in complex with the Fab portion of a previously identified neutralizing antibody (nAb) 1D5[36]. This antibody elicits a high neutralizing titer against CVA6 (Supplementary Fig. 8a, b), and also exhibits a high binding affinity for A-particles as revealed by an equilibrium dissociation constant ($K_D$) value of less than 10 nM (Supplementary Fig. 8c). CryoEM images of CVA6 A-particles incubated with excess Fab-1D5 clearly show particles with a halo of extra density compared to purified A-particles (Supplementary Fig. 2c). Two-dimensional (2D) classification of such images show well-

defined Fab density features that extend ~60 Å above the CVA6 surface (Supplementary Fig. 2d). A total of 12,067 particle images were extracted and subjected to 3D reconstruction yielding a final density map at an estimated resolution of 3.8 Å (FSC = 0.143)[37] (Supplementary Fig. 3g–i and Supplementary Movie 2). The density map shows that five Fab molecules encircle the topmost edge of the 'cooling tower' at each 5-fold vertex, and therefore a total of 60 Fab molecules can bind to each capsid (Fig. 5a, b and Supplementary Movie 2). The variable domains (directly contacting the capsid epitope) of the bound Fab molecules exhibit density approximately as strong as that of the capsid shell, which supports the notion that the Fab sites on the capsid are nearly fully occupied. Also, as expected, owing to the flexible connection between the constant and variable domain in the Fab, the constant domains exhibit weaker density compared to the capsid shell and the Fab variable domains (Fig. 5b and Supplementary Fig. 9a).

The atomic model of the CVA6 A-particle-1D5 immune-complex provides detailed views of several important features including the Fab paratope, the virus epitope, the A-particle-Fab interactions, and also Fab-Fab interactions (Fig. 5c–e, Supplementary Fig. 9b, Supplementary Table 3, and Supplementary Movie 3). Identification of a Fab interaction footprint site on the virus surface was performed by the PISA program[38], and the interaction interface of each Fab covers an area of 675 Å$^2$ (Supplementary Fig. 10). The interaction interface of Fab 1D5 involves five out of the six total complementary determining regions (CDR) and one framework region (FR) in the light chain: CDRL1, 2 and 3 and CDRH2, 3 and the FRL3 (between CDRL2 and CDRL3) (Fig. 5d, e). The A-particle-1D5 interaction interface includes surface loops on two VP1 proteins from two adjacent protomers. Each Fab interacts with the BC, EF and HI loops of the nearest VP1 subunit and the DE loop of the counter-clockwise-related (viewed from outside the complex) VP1. This Fab-VP1 interaction includes an array of eight hydrogen bonds and van der Waals interactions. The BC, EF, HI loops from one VP1 and DE loop from an adjacent VP1 all hydrogen bond (H-bond) to a single Fab (Fig. 5f–h and Supplementary Movie 3). Aside from the above mentioned hydrogen bonds, residues on the VP1 EF loop contribute the majority of van der Waals interactions between CDRH and VP1.

The five Fab molecules aligned at each vertex H-bond to each other via pairs of adjacent light chains (Fig. 5i). This H-bond forms between the first residue of the light chain (ASP1) on one Fab and TYR66, which lays 3 Å away on the light chain of an adjacent Fab (anti-clockwise, viewed from outside the complex). This network of five H-bonds acts to tether the otherwise flexible N-termini of the light chains and forms a stable, wreath-like ring of five Fabs. Notably, as we mentioned before, each Fab molecule covers a footprint on epitope with a surface area smaller than an average value of ~1000 Å$^2$ [39], which indicate an inferior binding affinity between CVA6 A-particle and 1D5. However, such wreath-like five-Fabs coalition covers a huge footprint (an area of 3375 Å) on the A-particle capsid surface, which is consistent with the high binding affinity of 1D5. This might explain why the CVA6 A-particle-1D5 is a rather stable immune-complex, which enables its structure to be determined at high resolution.

**1D5 prevents virus attachment to cell.** To further explore the mechanism by which 1D5 neutralizes CVA6, we examined whether it could inhibit virus-cellular binding or interrupt the post-attachment step of infection. A previously reported poly-merase chain reaction with reverse transcription (RT-PCR) assay was used to measure the number of virus particles bound to host cells[39]. When pre-mixed with CVA6 virus particles, 1D5 reduced

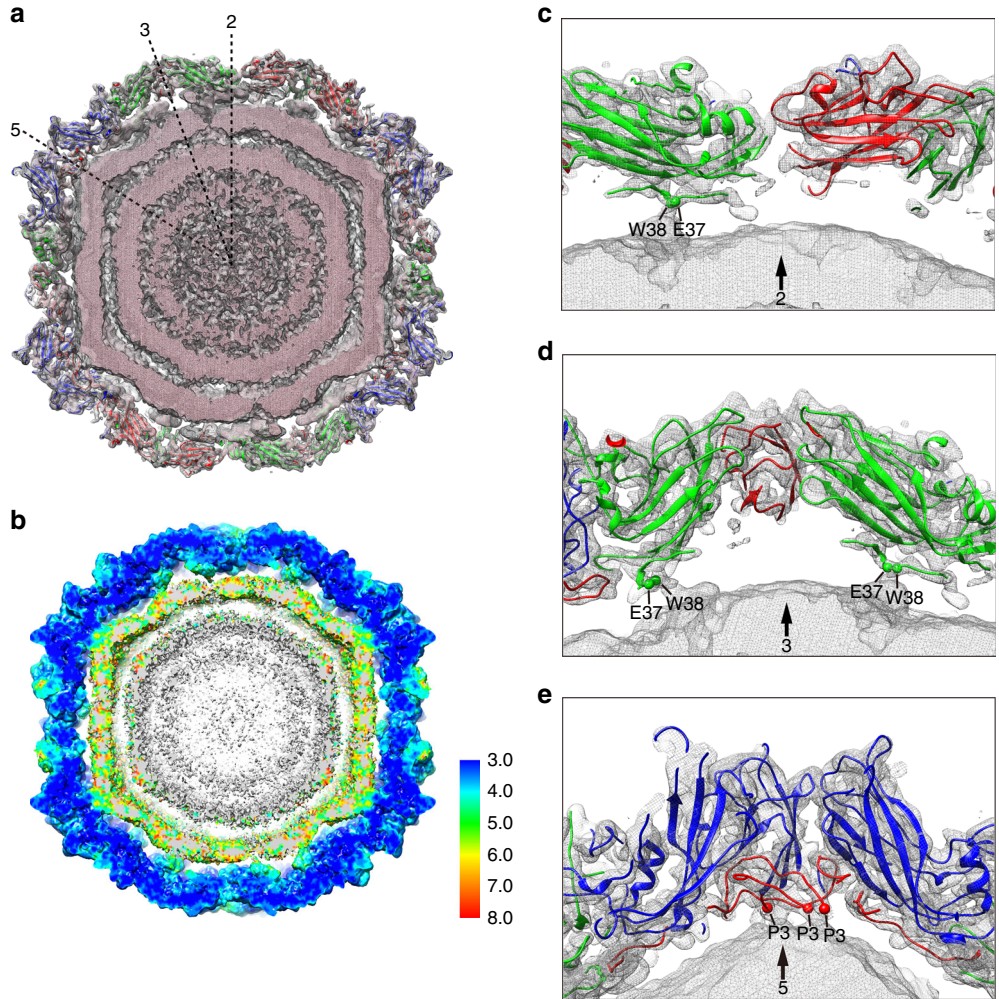

**Fig. 4** Capsid-RNA interactions in the CVA6 A-particle. **a** Central slab (25 Å thickness) of the CVA6 A-particle density map (*light pink* meshed density) and corresponding atomic model (ribbon diagrams) viewed along an icosahedral 2-fold axis. Capsid-RNA interactions can be seen beneath the VP2 and VP3 subunits near the 5-fold axis. 2-, 3- and 5-fold icosahedral symmetry axes are labeled with dashed lines. **b** The details of local resolutions related to capsid-RNA interactions. **c–e** Close-up views at the capsid-RNA interface are shown for regions near the 2-fold **c**, 3-fold **d** and 5-fold **e** axes. The N-termini of all VP2 subunits contact the genome in disordered regions of the density map, which precluded building an atomic model. Therefore, only the nearest (close to interaction interface) and identifiable residues are labeled (*spheres*). Strong, continuous densities are observed between VP3 and the RNA genome at the 5-fold axes **e**. Contacts are mediated by residue P3 (*red spheres*) at the N-termini of VP3

significantly the number of virus particles bound to the cell surface with the inhibition level correlated directly with 1D5 concentration (Fig. 6a). Furthermore, similar inhibition by 1D5 was found when virus particles were allowed to bind to cells before 1D5 was added, which indicates that 1D5 has much stronger affinity to CVA6 particles than cell does (Fig. 6b). 1D5 seemed to be more efficient in blocking CVA6 from attaching to cell surface than in competing against cell surface for occupying the binding sites on virus particles.

At least up to the resolution (3.8 Å) of our cryoEM map, Fab-1D5 binding induces no significant structural rearrangement of the CVA6 capsid as revealed by their superposition of the protomers in the CVA6 A-particle-1D5 complex and in A-particle and also reflected by the r.m.s.d. value of 0.4 Å between the Cα atoms of the two structures (Supplementary Fig. 9c and Supplementary Table 2). Because the ring of five Fabs that encircle the tip of each 5-fold vertex are connected by intermolecular H-bonds, this may act to stabilize the capsid and prevent conformation changes required for genome release. The fluorescence-based thermal stability assays appear consistent with this hypothesis given that CVA6 immune-complexes melt at a

1 °C higher temperature than virus particles in the absence of Fab or Ab binding (Fig. 6c).

The epitope recognized by the 1D5 was also shown to be an immune-dominant antigenic site. Antisera against CVA6 procapsid, untreated A-particle or heat inactivated A-particles were prepared and used to probe competitive binding of 1D5. All three antisera could similarly and efficiently inhibit binding of 1D5 to CVA6 procapsid (58.7%, 50.2% and 57.9%) or A-particle (62.9, 41.6 and 58.0%) (Fig. 6d). The above results indicate that 1D5 binds to an immune-dominant site on CVA6.

## Discussion

Picornaviruses primarily exist non-infectious, empty procapsids and infectious, mature virions in infected host cells. Upon attachment to specific receptors on the host cell surface during the initial stage of the infection in vivo, virions expand and transform into A-particles[14, 26]. Such a transformation can also be induced in vitro when virions are treated with heat, low pH or allowed to interact with receptor[15, 24, 26, 40, 41]. To date, A-particles have been characterized as necessary entry intermediates during infection. The atomic structure of A-particle for

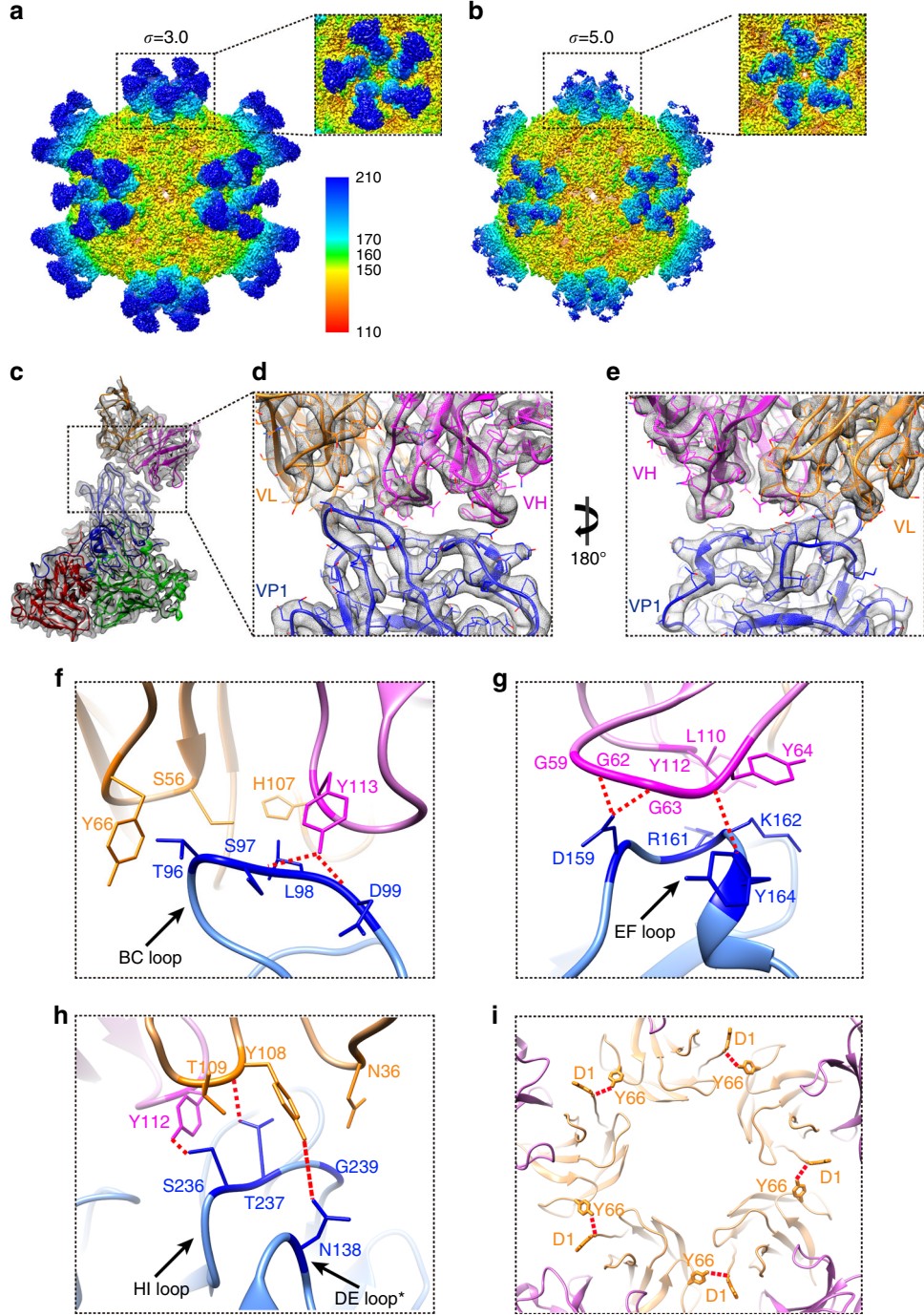

**Fig. 5** CryoEM map and atomic model of CVA6 A-particle complexed with Fab-1D5. **a, b** The isosurface views (radially colored) along a 2-fold axis of the CVA6-Fab cryoEM density map displayed at two different contour levels: $\sigma = 3.0$ **a** and $\sigma = 5.0$ **b**. The variable (*light blue*) and constant (*dark blue*) domains of each Fab can be seen in **a**, but most of the constant domain disappears at the higher threshold level **b** owing to the flexibility at the elbow region of the Fab. **c** Segmented immune-complex density map (*gray*) and fitted atomic models (ribbon diagrams) of three capsid proteins and the variable domain (light chain, *orange*; heavy chain, *magenta*) of the Fab-1D5. **d, e** Close-up view of the interaction interface involving five of six CDRs on the Fab and four surface loops of VP1. **f–h** Expanded views of the virus-Fab interaction interface close to the BC **f**, EF **g**, and HI loops **h** of VP1. Potential H-bonds are marked by red dashed lines. Residue Y108 in the Fab light chain can form a H-bond with T237 on EF loop and also N138 on the DE loop (black asterisk) of the adjacent VP1 capsid protein **h**. **i** Inside view of the five Fab molecules shows the potential H-bond formed between D1 and Y66 from two adjacent light chains

CVA16 has been reported previously, but the conversion occurred without any treatment in only one batch of multiple isolations and not yet be duplicated[14]. In this study, however, CVA6 procapsids and A-particles can be obtained by routine methods of continuous, sucrose density gradient ultracentrifugation. CVA6 A-particles isolated and purified in this manner exhibit the biochemical composition and structural features similar to the A-particles of other picornaviruses. Indeed, these CVA6 A-particles have a sedimentation coefficient of 128S, which is similar to 135S of A-particles for most picornavirions. Also, like A-particles, the VP4 capsid protein is absent in these CVA6 A-particles and they have an elongated 2f-channel and collapsed pocket factor binding site. However, unlike the elusive A-particles, CVA6 A-particles are

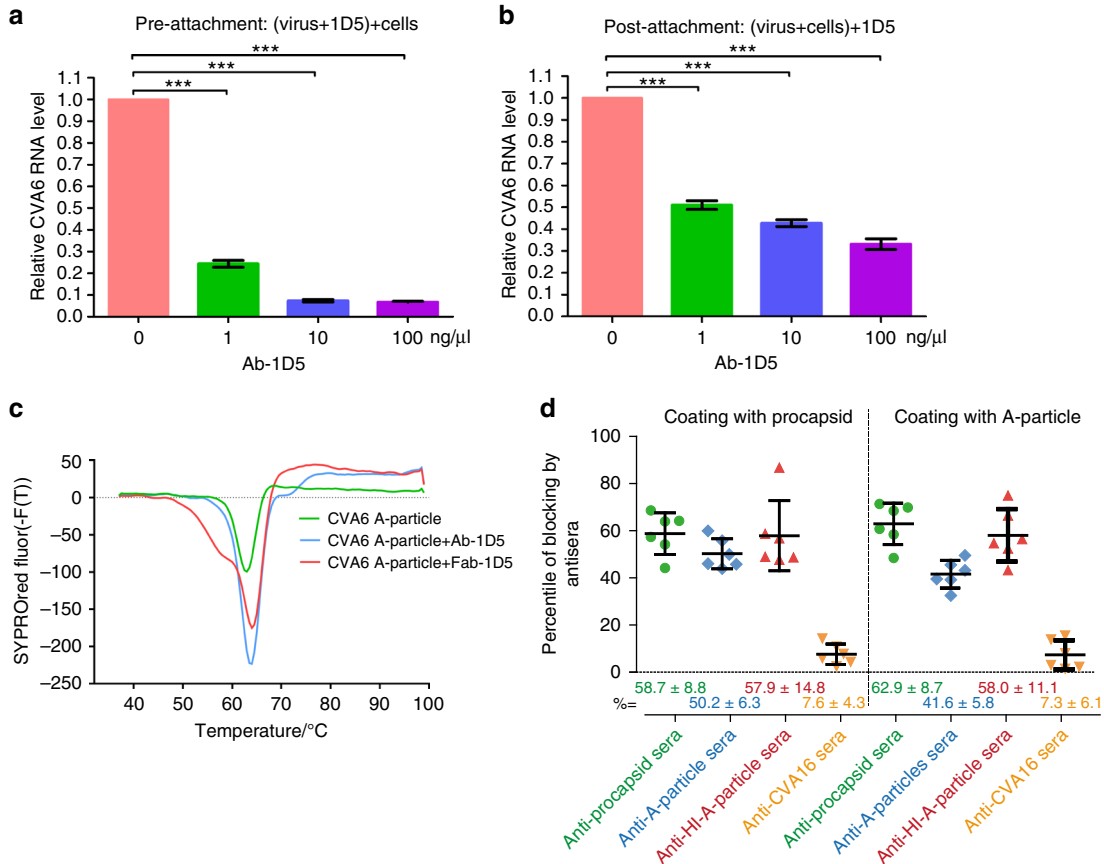

**Fig. 6** Potential mechanism of 1D5-mediated neutralization of CVA6 A-particle. **a, b** The number of cell-bound CVA6 virus particles bears an inverse relationship to Ab-1D5 concentration. The number of cell-bound virus particles was determined by detecting the amount of genomic RNA ($AMT_{RNA}$) by means of RT-PCR, and, as a control, $AMT_{RNA}$ was detected in the mixture of particles and RD cells in the absence of Ab-1D5. The ratio ($AMT_{RNA}$ in test/ $AMT_{RNA}$ in control) gradually decreased as a function of increasing Ab-1D5 concentrations either before **a** or after **b** virus particles were plated on RD cells. Values along the Y-axis are expressed as mean ± s.d. Statistical significance was analyzed by the $t$-test (***$P < 0.001$). **c** Structural stability of the A-particle complexed with either Ab-1D5 or Fab-1D5. Exposure of genomic RNA due to structural instability induced by heat A-particle treatment of the immune-complex was indicated by the value of first derivative of the fluorescence. **d** Surface loops of VP1 act as an immune dominant epitope of the CVA6 A-particle. Binding of 1D5 to procapsid (left of *vertical dashed line*) or A-particle (right side) is significantly blocked by antisera against procapsids, A-particles, or HI-A-particles, but marginally by antisera against CVA16 (mean ± s.d.)

biochemically and structurally stable both in vivo and in vitro while remaining infectious and immunologically active.

Specific receptors have been identified for many picornaviruses[31, 32], but the receptor for CVA6 has not yet to be identified. Picornavirus receptors typically bind to the highly exposed surface loops of the capsid proteins, which also often serve as the viral neutralizing epitopes. The binding sites of the poliovirus or rhinovirus receptors have been identified on the BC or HI loops of VP1[42–44]. EV71 and CVA16 both employ SCARB2 and PSGL-1 as their cellular receptors[31, 32]. For EV71, SCARB2 has been shown to bind in the canyon or 'pocket' region on the EF loop of VP1[45]. Hence, VP1 surface loops appear to be a preferred receptor-binding site in many picornaviruses. Obviously, comparisons of the CVA6, EV71 and CVA16 capsid structures show that the prominent differences among these structures map to the BC, DE, EF and HI loops of VP1. Therefore, distinct arrangements of these four loops among different picornavirus species creates numerous unique capsid surfaces and likely explains why CVA6 does not share the same receptors with EV71 or CVA16. The atomic model of the CVA6 A-particle-1D5 complex clearly demonstrates that these loops also serve as the viral neutralizing epitope, and 1D5 has the ability to disrupt binding or attachment of CVA6 to host cells. This suggests that the CVA6 receptor-binding site may be at or close to the VP1 surface loops.

Furthermore, the VP1 loops would appear to be excellent targets for designing specific antivirals against CVA6.

It is well established that mature picornavirus particles undergo significant structural expansion during the uncoating process as they transform into A-particles that release the genomic RNA into host cells[12, 14, 15]. Here, the CVA6 A-particle examined in this study, unlike the classical, mature picornavirion and instead more like a stable uncoating intermediate, may require only minimal conformational change in its capsid to enable release of the genomic RNA. Hence, the RNA in CVA6 A-particles may be more primed for release compared to the packaged RNA in mature virions of other known picornaviruses. In fact, the capsid-RNA interactions revealed in our CVA6 cryoEM density maps might provide some clues as to where the genome exits the capsid as it is released into host cells. As our results show, the N-terminus of VP3 in the CVA6 A-particle forms strong interactions with the genome at the icosahedral 5-fold vertices, which is unique among all the known enteroviruses[14, 15, 26]. In turn, the binding of five Fab-1D5 molecules at each CVA6 vertex can not only neutralize the CVA6 A-particle but it might also stabilize the capsid and thereby restrict conformational changes necessary to allow RNA release. We also noted that such VP3-RNA interaction was observed in A-particles of rhinovirus captured at its near-physiological conditions by acidification of mature

rhinoviruses (as occurs in the endosome) as well[28]. However, unlike these unstable rhinovirus A-particles, CVA6 A-particles are readily harvested in vivo and remain stable in buffer (pH 5.5) that mimics the optimal physiological environment of the human gut. This might be one of the reasons that CVA6 adopts A-particle, rather than mature virus, as its stable being in its life cycle. In this study we have shown that CVA6 exhibits several unique properties that identify it as an anomalous member of the family *Picornaviridae*.

The cryoEM structures of the CVA6 procapsid, A-particle, and A-particle-1D5 immune-complex we report here reveal atomic resolution details about molecular determinants in CVA6 related to virus neutralization, cell binding, and RNA release. These details also suggest that procapsids and A-particles (heat-treated) might serve as excellent vaccine candidates against HFMD. Finally, studies like this one, in which the detailed interaction interface between a virus and antibody is directly visualized, highlight the prediction by Earl and Subramaniam[46]: "cryoEM is likely to increasingly become a routine tool to aid rapid vaccine design, especially icosahedrally symmetric viruses".

## Methods

**Virus production and purification**. CVA6 strain TW-2007-00141 (GenBank accession no. KR706309 that was first isolated in 2007) in Taiwan was grown in RD cells at a multiplicity of infection (MOI) of 0.1. Virus was harvested 3 days post infection, centrifuged to remove cell debris, ultrafiltered. The virus supernatant was mixed with 50% polyethylene glycol 8000 (PEG 8000) and 2 M NaCl-phosphate-buffered saline (PBS) (pH 7.4) to final concentrations of 8% and 300 mM, respectively, and stirred overnight at 4 °C. After centrifugation and removing the pellet, the CVA6 virus particles were loaded onto a 15–45% (w/v) sucrose density gradient and centrifuged at 153,900 *g* for 4.5 h in a Beckman SW41 rotor at 4 °C. Two sets of fractions were independently dialyzed against PBS buffer and further concentrated. The quantity of virus particles were then estimated by its UV absorption level at a wavelength of 260 nm using UV spectroscopy. The protein composition was analyzed with SDS–PAGE. The concentration and homogeneity of the particles were also examined by negative-stain electron microscopy.

**PaSTRy assay**. Thermofluor experiments[47] were performed with a MX3005p RT-PCR instrument (Agilent/Stratagene). Fluorescent probe SYTO9 and SYPRORed (both from Invitrogen) were used to detect the presence of single-stranded RNA and exposed hydrophobic regions of capsid proteins respectively. Multiple reaction mixtures, each with a total volume of 50 μl, containing 1.0 μg of virus particles, 5 μM SYTO9 and 3 × SYPRORed, but with different a pH scale (ranging 5.5 to 8.5, 0.5 interval), were set up in thin-walled PCR plates (from Agilent). The fluorescence level was recorded in triplicate at 0.5 °C intervals from 25 to 99 °C. In addition, a similar thermal stability assay was also performed on virus-Fab or virus-Ab immune complexes with 1.0 μg of CVA6 particles pre-incubated with either antibody or Fab (with a final antibody/fab concentration of 50 μg ml$^{-1}$) at 37 °C for 1 h. The temperatures at which RNA was released (Tr) and at which particles melted temperature (Tm) were recorded as the minimums of the negative first derivative of the RNA exposure and protein denaturation curves, respectively.

**Vaccine preparation and immunization of mice**. The purified CVA6 A-particles were inactivated by heating at 56 °C for 30 min (heating causes RNA release, producing non-infectious particles[48] and their immunogenicity was evaluated in mice. Four groups of mice (*n* = 6 per group) were respectively immunized (two doses, 3 weeks apart) with aluminum adjuvant (control), CVA6 procapsids, CVA6 A-particles, or pre-heated CVA6 A-particles. Sera were inactivated by incubation at 56 °C for 30 min, and stored at −20 °C for ELISA and neutralization assays.

**In vitro neutralization assay**. RD cell monolayers were diluted in MEM supplemented with 2% FBS and then seeded into 96-well plates (NUNC) (~10,000 cells per well). Different concentrations of mAbs, Fab fragments, or HI mouse sera were diluted in MEM by 2-fold serial dilutions ranging from 1:8 to 1:4,096, and each well was incubated in 1:1 volume ratio with infectious CVA6 strains (100 TCID$_{50}$) for 1 h at 37 °C. Each sample was then incubated with the prepared RD cells in 96 well plates at 37 °C and was inspected for the CPE phenomenon in infected cells for 3 days. The neutralization titers were the averages of the triplicates calculated based on the highest dilution in over 50% CPE.

**Pre- and post-attachment inhibition assays**. For pre-attachment inhibition assays, serially diluted 1D5 was incubated together with CVA6-141 virus ($5 \times 10^4$ TCID$_{50}$) for 1 h at 4 °C. The mixture was then added to $10^5$ RD cells which had been pre-seeded one day ahead in a 96-well plate and incubated for 1 h at 4 °C. The

infected cells were washed three times with cold PBS buffer to remove unbound virus. For the post-attachment inhibition assay, CVA6-141 virus was added to RD cells and incubated for 1 h at 4 °C. The virus-bound cells were washed twice with cold PBS buffer, mixed with serially diluted 1D5, incubated for 1 h at 4 °C and washed three times again. The amount of CVA6 was estimated by a one-step, quantitative RT-PCR as previously described[49]. In brief, RNA was extracted using the QIAamp Mini viral RNA Extraction Kit (Qiagen, Inc. Hilden, Germany) following the manufacturer's instructions. RT-PCR reaction was performed using the CFX96 Real-Time PCR Detection System (Bio-Rad). The total 50 μl reaction volume contained 0.5 μl each of 10 mM forward (5′-TACTTTGGGTGTCCGT GTTT-3′) and reverse primers (5′-TGGCCAATCCAATAGCTATATG-3′), 0.2 μM of the probe (5′-FAM- AYTGGCTGCTTATGGTGACRAT-BHQ1-3′), 3.2 mM of dNTP, 4 μl of 10 × buffer (Mg$^{2+}$ plus), 1 U of Taq HS DNA polymerase, 0.4 U of AMV reverse transcriptase and 5 μl of extracted RNA template. The thermal profile for RT-PCR was 15 min of reverse transcription at 50 °C; 10 min of denaturation at 95 °C; 40 cycles of 95 °C for 15 s, and 55 °C for 45 s. The analysis of relative levels of CVA6 RNA in different samples was performed by the comparative $2-\Delta\Delta CT$ method[50].

**Competitive ELISA**. The sera from mice immunized with CVA6 particles were used for the competitive ELISA assay. The sera from mice (at 1:100 dilution) were added to the wells coated with purified CVA6 particles in blocking solution (final volume: 50 μl per well), and the plates were incubated for 30 min at 37 °C. HRP-conjugated 1D5 (1:5000 dilution) was then added and incubated at 37 °C for 30 min. The OD value was converted to percentage inhibition (PI) using the following formula: PI (%) = 100−[(OD$_{sample}$/OD$_{control}$) × 100], where the OD of the control well represents the well containing only HRP-conjugated 1D5.

**CryoEM image acquisition**. Immune complex were prepared by mixing CVA6 A-particle (1.4 mg ml$^{-1}$) with 1.2-fold excess of Fab-1D5 (MW = ~50 kDa, 108 a.a. and 119 a.a. corresponding to light and heavy chains of the variable domain) fragment (equivalent to a molar ratio of 1:72), then incubated at 37 °C for 2 h. Aliquots (3 μl) of purified samples of procapsid, A-particle or immune complex were deposited onto glow discharged holey carbon Quantifoil Cu grid (R2/2, 200 mesh, Quantifoil Micro Tools) inside an FEI Mark IV Vitrobot at a humidity level of 100%. After 6s blotting, the grid was plunge-frozen into liquid ethane cooled by liquid nitrogen, and then examined under low-dose conditions at 300 kV with an FEI Tecnai F30 transmission electron microscope. All images were recorded on a Falcon II direct electron detector (seven-frame movie mode) with the defocus settings ranging between 1.5 and 3.0 μm underfocus and at a nominal magnification of 93,000 (corresponding to a pixel size of 1.128 Å). The total electron dose was set to 25 e$^-$ Å$^{-2}$ with an exposure time of 1 s. The FEI EPU automated data collection software was used for all data acquisition, and micrographs with excessive drift or astigmatism were discarded. A total of 312 micrographs for CVA6 procapsid, 203 micrographs for the A-particle and 1085 micrographs for the immune complex were selected for further image processing.

**CryoEM single particle 3D reconstruction**. Movie frames alignment and CTF estimation of each aligned micrograph were carried out with the program Motioncorr[51]. and Ctffind3[52]. A total of 23,172 CVA6 procapsid, 9993 CVA6 A-particles and 13,765 immune complex particles were manually boxed using the e2boxer.py routine in the EMAN2.1 package[53]. Initial 3D models of each of the three kinds of particle were generated with random model method using AUTO3DEM[54]. Two rounds of reference-free 2D classification, several rounds of unsupervised 3D classification and final 3D density map reconstruction were all performed with RELION 1.4[55]. A total of 10,749 selected procapsid images, 7152 selected A-particle images and 12,067 CVA6 immune complex particle images were included in final 3D reconstructions respectively. The resolutions of the final maps were estimated based on the gold-standard FSC curve with a cut-off at 0.143[37]. Local resolution variations were estimated using ResMap[56].

**Model building and refinement**. The atomic models of all three particles were manually built in Coot and then refined with Phenix[57, 58]. The crystal structure of the CVA16 virus protomer (PDB code 4JGY)[14] was used as a homology model and manually fitted into the segmented volume (including an asymmetric unit) of the final cryoEM map using Chimera[33], whereas a de novo modeling procedure was performed on the final cryoEM map of the CVA6 A-particles-1D5 complex. Atomic positions of residues were built and modified using the Coot software package and further refined using Phenix in real space to maximize the correlation coefficient between the EM density map and a calculated map based on the coordinates of all atoms. The above procedure was iteratively repeated five times to achieve the optimized atomic model of one asymmetric unit. Model statistics, including bond lengths, bond angles, all atom clashes, rotamer statistics, Ramachandran plot statistics, etc, were closely inspected with Coot during the whole process. Visualization, segmentation of density maps, and generation of animation movies were performed with Chimera[33]. Fab density in the difference maps were projected on a stereographic sphere using RIVEM[59]. The Fab 1D5-CVA6 interactions were analyzed using Chimera and the PISA[38] server (www.ebi.ac.uk/pdbe/pisa) with the donor to acceptor distances <4 Å for hydrogen bonding interactions.

**Cell lines and media and viruses**. Human embryonal RD cells (from American Type Culture Collection) were cultured in Minimal Essential Medium (GIBCO) supplemented with 10% FBS (GIBCO). The CVA6 strain was isolated from throat and anal swabs of HFMD patients in Taiwan. CVA6 stock was propagated in RD cells and the supernatant was harvested and stored at −80 °C. Virus was titrated by a plaque assay using RD cells. Virus samples were added to RD cells in 96-well plates and incubated at 37 °C for 7 days. Then the 50% tissue culture infectious doses (TCID$_{50}$) values were measured by determining CPE and calculated according to the Behrens-Kärber method.

**SDS–PAGE**. The purified CVA6 particles were analyzed by SDS–PAGE. Virus particles were mixed with the 4 × NuPAGE LDS sample buffer and 10 × reducing agent (Invitrogen). Sample mixtures were heated at 70 °C for 10 min before loaded onto a NuPAGE 4-12% Bis-Tris Gel (Invitrogen).

**Negative staining electron microscopy**. The purified CVA6 particles solution at a concentration of 0.1 mg ml$^{-1}$ were deposited onto a carbon-coated grid for 1 min and the excessive solution was removed with filter paper. The grid was washed twice with double distilled water and then immediately negatively stained for 30 s with freshly filtered 2% phosphotungstic acid (pH 6.4). Grids were examined with the FEI Tecnai T12 TEM at an accelerating voltage of 120 kV and photographed at a magnification of 25,000.

**Analytical ultra-centrifugation**. The homogeneity and molecular mass of the CVA6 viral particles were estimated by the value of sedimentation velocity (SV) achieved with AUC. CVA6 viral particles were first diluted to 0.2 mg ml$^{-1}$ in PBS buffer, AUC analysis was performed at 4 °C with Beckman XL-A analytical ultracentrifuge, equipped with absorbance optics and an An60-Ti rotor (22,700 g). A total of 150 scans for each sample were recorded and the sedimentation coefficient and f/f0 were obtained with the c(s) method[60] using the Sedfit software (kindly provided by Dr P. Schuck, National Institutes of Health, http://www.analyticalultracentrifugation.com/).

**SPR**. The binding of 1D5 to CVA6 A-particles was analyzed by SPR using a BIAcore 3000 biosensor (GE Healthcare Life Sciences). The affinity measurements of 1D5 binding with CVA6 A-particles were initiated by passing HBS (10 mM HEPES, pH 7.4 and 150 mM NaCl) over the sensor surface for 100 s at 5 µl min$^{-1}$, followed by injection of 5 µg ml$^{-1}$ of 1D5 at 30 µl min$^{-1}$ for 2 min. Serially diluted A-particles were then injected at concentrations of 0, 0.172, 0.345, 0.69, 1.38 and 2.76 nM, at a flow rate of 30 µl min$^{-1}$ for 3 min, and then allowed to dissociate over 2 min. Binding curve at the zero concentration of particle was subtracted as a blank from each experimental curve. Data were analyzed using BIA evaluation 4.1 software. Kinetic constants, $k_a$ and $k_d$, were estimated by global fitting analysis of the association/dissociation curves to the 1:1 Langmuir interaction model.

**Preparation of Ab-1D5 and Fab-1D5**. To obtain the anti-CVA6 monoclonal antibody (1D5), mice were immunized with CVA6-00141 emulsified in Freund's adjuvant. Two booster immunizations were performed on days 14 and 28. After final boost, fusion of splenocytes with Sp2/0Ag-14 myeloma cells were performed. Then the hybridoma supernatants were screened by indirect ELISA and neutralizing test against CVA6. Purified 1D5 was got from mouse ascitic fluid by purifying with protein A column. The 1D5 was conjugated to horseradish peroxidase (HRP) by the NaIO$_4$ oxidation method and stored at −20 °C.

The RNA was isolated from 1D5 hybridoma cells and converted to cDNA by reverse transcription. Then, the variable regions of the heavy chain and light chain of 1D5 were amplified by PCR for sequence determination. The Fab-1D5 fragment was digested with papain at a weight ratio 800:1 in 20 mM phosphate buffer (pH 7.0) containing 30 mM L-Cys and 50 mM EDTA for 10 h at 37 °C before adding 30 mM iodoacetamide to stop the reaction. The resulting Fab fragment was then purified using DEAE-5PW (TOSOH).

**Binding ELISA**. The 96-well plates were coated with 50 ng/well of purified CVA6 procapsids and A-particles (0.5 µg ml$^{-1}$ in PBS buffer) or CVA6-infected RD cell lysates and incubated at 4 °C for 12 h. The plate was then washed with PBS containing 0.05% Tween 20 and 1% bovine serum albumin in PBS for 2 h at 37 °C to minimize the nonspecific binding. The antibodies were then added at various concentrations. The serum with 10-fold dilution series (the first dilution was 100-fold), were added to the wells and incubated for 30 min at 37 °C. The horseradish peroxidase-conjugated goat anti-mouse (GAM-HRP) IgG antibody/well in a 1:5,000 dilution was then added and incubated for 30 min at 37 °C as well. After color development, absorption was measured at A450/620.

**Data availability**. Atomic coordinates of CVA6 procapsid, A-particle and A-particle-1D5 complex have been submitted to the Protein Data Bank with accession numbers 5XS5, 5XS4 and 5XS7, respectively. The cryo-EM density map has been deposited with the Electron Microscopy Data Bank 6752, 6751 and 6757, respectively. Additional data that support the findings of this study are available from the corresponding authors upon reasonable request.

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

## Acknowledgements

This work was supported by a grant from the National Natural Science Foundation of China (No. 31670933 and 81401669), the National Science and Technology Major Projects for Major New Drugs Innovation and Development (No. 2017ZX09101005-005-003), the National Science and Technology Major Project of Infectious Diseases (No. 2017ZX10304402-002-003) and the Natural Science Foundation of Fujian Province (No. 2015J05073). This work was also supported in part by funding to T.S.B. from the National Institutes of Health (Grant R37-GM33050). The funders had no role in the study design, data collection and analysis, decision to publish, or preparation of the manuscript.

## Author contributions

N.X., T.C., X.Y., Z.H.Z., S.L., L.X. and Q.Z. designed experiments. L.X., Y.W., Y.L. and R.Z. cultured cells and purified virus samples. L.X., M.H., Y.W., Y.L., R.Z., S.L., Z.H., J.J., L.Y., W.W., and W.H. performed research. L.X., Q.Z., M.H., H.Y., Q.H., X.Y. and Z.L. analyzed data and T.C., X.Y., Z.H.Z., S.L., T.S.B., L.X. and Q.Z. finalized the manuscript. J.Z., T.S.B., S.L., Z.H.Z., X.Y., T.C. and N.X. participated in discussion and interpretation of the results. All authors contributed to experimental design; Z.H.Z. and T.S.B. supervised the project.

## Additional information

**Competing interests:** The authors declare no competing financial interests.

