## [Peer Review File · Nature Communications]

Reviewers' comments:

Reviewer #1 (Remarks to the Author):

The manuscript "Atomic structures of Coxsackievirus A6 and its complex with a neutralizing antibody reveal molecular determinants for vaccine development against hand, foot and mouth disease" by Xu and co-authors reports the study of Coxsackie A6 virus (CVA6) using combined approaches: biochemistry, cryo-EM structural analysis, and immunization experiments. This virus is a member of the human Enterovirus species A. The authors report three structures: the CVA6 procapsid, mature CVA6 virion, and the virion-1D5 immunocomplex at nearly atomic resolution ($\sim 3.5 \text{ \AA}$). Comparison of these structures demonstrates that loops (BC, DE, EF, and HI) in capsid protein VP1 of CVA6 are involved into interactions with the host cells. This was confirmed by results of biochemical and immune experiments when binding of Fabs induces virus neutralization. The outcome implies that procapsids and heat-treated virions can be used as vaccine candidates to fight HFMD. Structures delivered detailed information on interaction interface between a virus and antibody that is likely suggest means for effective rapid vaccine design.

Images of viral particles were collected on FEI Tecnai F30 cryo microscope equipped with Falcon II camera using EPU automated data collection system. The authors have used Relion software and $\sim 10,000$ images for the procapsid, $\sim 7,000$ for the virions, and $\sim 12,000$ for the virion/Fab complex. The authors have obtained three structures at resolutions allowing to identify residues forming polypeptide chains. Comparison of sequences of capsid proteins from CVA6, EV71, CB3, and EVD68 viruses, their secondary elements, and available atomic structures reveals that all structures are very similar. That is not surprising at such high conservation in polypeptide chain where only specific residues in the VP1 proteins responsible for the neutralizing antigenic epitopes are slightly different which may explain specificity of the viruses. The research presented indicates very nicely how a combination of biochemistry, structural analysis and immunization experiments could lead to the practical applications in medicine.

The authors have to take in account the following:

- Abstract, line 29. Absence and presence of "RNA interactions": It is unclear what the authors mean. If they mean interactions with capsid proteins it should be indicated.

See comments below related to map interpretations.

- Page 4, lines 78-79. I would recommend to remove this sentence, since the authors had a chance to resolve this "mystery": they had a sample where they were able to find both empty and filled with genome particles of CVA6 (supl figure 2b). The authors had a wonderful opportunity to find differences between these particles but that was not done. Therefore the sentence does not make sense. Why it was not done it is another question.

- Page 4, lines 81-82. The loops are representing a small percentage of the entire set of sequences of capsid proteins, so it is hardly expected that viruses will be much different in structures and mode of infection. However, specificity to the host cells should be reflected in differences of a few amino acids which are shown by sequence alignments.

- Page 5, lines 100 and 104. It seems that the authors have indicated that protein V4 was not present in the capsids. Explanation should be given in this part related to the both procapsids and mature capsids. Or at least the authors have to prove that V4 was in procapsids.

- Page 6, line 124. It would be helpful for a reader if the authors will explain why heating inactivates the virus, otherwise it is confusing.

- Page 7, line 134. "low-dose cryo EM methods" -> please rephrase. In EM we do not have different "low-dose methods". A method is the only one that we use, but we have slightly different technical implementations so it should be the "low-dose imaging" or the "low-dose technique"

- Page 7, line 240. It seems that authors have one average diameter of the capsid but distances between proteins located on 2, 3, and 5-fold axes are vary. It is still not clear between what and what measurements were done.

- Page 7, line 145. I am not sure that RMSD values are very informative for assessment of differences between EM maps. How the authors can prove that these variations above deviations

in their measurements? This is an assessment of variations in the pseudo atomic models but not in EM maps. If another person will do the flexible fit in the same map, what would the RMSD be between two fittings? It is again related to the "golden rule".

- Page 7, line 147. The authors like very much the expression "Of note". I am not sure that it is informative and nearly in all 5-6 sentences within this MS it can be removed without changing the essence of sentences.
- Page 7, line 147. It is unclear to me what the authors implying by saying that VP4 was not seen due to its flexibility. It may be so, but do they have any idea where it can be located? Where to look for it? Did they prove that it was in procapsids?
 - Page 7, line 151-152, It would be good if the authors will rephrase the sentence. It sounds a bit confusing: "... several equivalent or similar regions devoid of residues because corresponding regions of density in the cryoEM maps were smeared...". I think a situation was reversed: the authors were not able to allocated position of the residues of the loops (that are present in real proteins) due to their flexibility and low local resolution in maps.
 - The authors are talking a lot of flexibility of certain areas in capsid proteins, but why not to show the local resolution of these areas? Relion can do that without any extra efforts. That has to be shown on several occasions where the authors are talking of the specific interactions between protein/protein and RNA/protein.
 - Page 7, line 154, It is not a good expression: "... both particles contain a segment of disorder in the AB loop ...". Should be rephrased.
 - Page 8. line 160, Sentence "...it has a strikingly similar capsid structure.." why the authors think it is "strikingly" if the protein sequences are nearly identical except loops and very ends of C- and T- termini? See the supl. figure 5.
 - Page 8, line 173, Not sure that this is the correct expression: "...show loss of ordered residues..." It seems that the authors again were not able to do the fitting and to make an atomic model, since the EM densities were not well defined... but residues were not "lost".
 - Page 9, line 181, RMSD values do not say anything bad or good. It is not much information here. Again just an indication of flexibility.
 - Pages 9-10, lines 188-202. The section related to Capsid /RNA interaction is very confusing. The authors show maps at extremely low threshold and a very thick slab, so one can put anything into these bulky areas of density. It is puzzling how the individual residues of VP2 that interact with genome were identified? Especially when the authors write correctly that there is a mismatch between the capsid proteins and RNA within the capsid: capsid proteins have symmetrical arrangement while genome is asymmetrical. Demonstration of the local resolution here will be helpful since the authors claim exact amino acids of the chain that are in the contact with RNA, while in the pictures a resolution seems to be hardly 10 Å.
 - Page 10, line 206. Sizes of the Fab used should be given both as M.m, and length in AA. That will help to asses quickly changes in sizes and its visibility on cryo images
 - Page 10, line 211. The authors write that densities corresponding to the Fab are well defined. Possibly yes..., however, it does not look very convincing in the suppl figure 2c. The authors did not provide any information on how the classes were obtained, how many images are in each class. Although it is natural to assume that the classes were obtained in Relion, then the classes would comprise up to several hundreds of images; so it would be risky to say that densities that protrude from the capsid in projections verify full occupancy of binding sites with the Fabs. Moreover, the densities seen as protrusion around the capsids are not so strong as densities corresponding to the capsids. So there are some doubts that all binding sites on the surface of the capsids are occupied by Fabs
 - Page 10, line 214. It is not very usual expression "... an estimated resolution limit..". Typically we are talking of a resolution of a structure but not of the resolution limit in the structure. The resolution can be limited by different factors.
 - Page 10, lines 217-220. The authors did not provide definitions of "constant", "variable", rigid, and flexible domains. It seems that sometimes they are mixed up themselves with flexibility and variability, but variable domains have "approximately" the same densities as the capsid. The authors have to make clear what they want to say.
 - Page 10, line 219. Please rephrase: "...the Fab sites on the capsid are saturated with high

binding affinity...". Should it be? : "...To assess saturation of higher-affinity binding sites ,,, experiment were .performed..." or whatever was done by authors to assess the occupancy of Fabs. May be the authors had something different in their minds.

- Page 11, line 226. "... Each Fab contacts a footprint.." Should it be "...identification of a Fab interaction footprint site on an virus surface was done by..."
- Page 11, line 237. The authors are writing about H-bonds between light chains of Fabs. Do they really see H-bonds? It would be good to see these areas of maps and indications of local resolution.
- Page 12, line 244. Here is very unusual sentence: "...such wreath-like Fab coalition covers a huge footprint...". It should be rephrased.
- Page 12, line 246, The authors write that "a well saturated and rigid immune-complex, which is proved to be advantageous for the atomic resolution solution" ... The sentence has to be rephrased. Even more, it contradicts to the sentence on the page 10 (line 220) where constant domains have weak densities and therefore were not resolved,
 - Page 12, lines 252-253. It is not very clear if the full occupancy by Fabs on capsids is required to prevent binding of viral particles to the host cells. Possibly not.
- Page 12, line 264. RMSD numbers do not provide any Info on the similarity between the structures, it is a level of similarity between atomic models and possibly may reflect only the accuracy of the fitting within a given map at a given resolution. The lower resolution -> more variations in the fitting.
- Page 14, lines 293-301. The authors are discussing the variation in positions of loops and claiming that there are significant (prominent?) differences. So again there is some contradiction: structures are nearly the same that have very low RMSD, nonetheless there are significant differences in loop locations. How are big the differences? How they were assessed. The amplitude of variations should be shown in figures. The authors are saying that the loops are involved in the interactions with receptors, but on the other side their sequences make the loops specific at recognition of receptors of the host cells. What else is changing in the loops in different viruses apart of their positions?
 - Page 15, lines 314-320. Release of RNA. The hypothesis is very confusing and not properly explained. The authors write that RNA has strong interactions with 5-fold vertices of the capsid; and therefore release of RNA can take place via 5-fold vertices. It does not sound as a logical hypothesis. The Fabs may block or arrest release of RNA from capsids as it was demonstrated by the biochemical experiments, In these type of viruses with T=3 symmetry Fabs have only this option since the binding sites are located around 5-fold axes. However how genome will be released remains unclear.
- Page 20, line 431 "Atomic positions of residues were built and adjusted inside Coot". The sentence has to be rephrased... I hope that is was not done "inside Coot" but using .a software package Coot. I did not find the reference for Coot while Chimera has the reference.
 - Figure 2. Panels a,b,and c are too small. Possibly it makes sense to show a part of the capsid and make a link with the asymmetrical subunit. Panel d should be moved to Figure 3 and combined with figure3 e -: all three models that have to be fitted into this asymmetrical unit and amplitudes of loop displacements have to be indicated.
 - Figure 3. Should be partly combined with Figure 2. See comments above.
 - Figure 4. A threshold used to show a surface of the 3D map is extremely low and does not indicate a resolution at which some amino acids can be resolved. The authors have claimed resolution $\sim 3.5 \text{ \AA}$, beta strands have to be clearly resolved, but in this figure it is definitely not the case. The authors indicating specific residues those are supposed to interact with RNA. But in this map they are not seen. It seems that these AA were taken from other structures. The AA are not shown in the "ball-and stick-mode" but represented as ball and sticks. The authors have mixed up letters used for designation of panels,
 - Figure 5. "Iso-contoured views" -> should be views of an isosurface. At which threshold they are shown? "...the constant domain has disappeared..." ? It has not, it was not resolved. The explanation of Fab organization has to be given in the main text. The domains have to be labelled in the figure. What is a difference between "close-up views" and "magnified in views"?
- Suppl Figure 2. Panel d- more information should be provided on classes and how they have

been obtained. Images look like they are harshly coarsened projections of a 3D reconstruction of the virion/Fab complex: they have very good mirror or close to 2-fold rotational symmetries. It is a bit strange. However the densities of Fabs do not say much about the Fab occupancy.

- Suppl Figure 3. This is the most puzzling figure. Panels a, b, and c represent the Fourier Shell correlation functions and they are strange: they are cropped from the bottom. The curve behavior indicates that FSC values became negative after crossing X axis which is very strange. That can take place if the maps in comparison have the opposite contrast at high frequencies? It means that the authors have applied a bit strange filter on maps, or something else went wrong during masking of 3D maps. The FSC should be shown in full without any truncations. Panels b, d, and f - > the local resolution maps are too small, they are tightly masked and show only surfaces at the same local resolution. What was the threshold used to show isosurfaces? The authors have to show cut-away central views of reconstructions and possibly only one quarter, so magnification will be sufficient to see variation in resolution.

- It would be helpful to see (in supplementary materials) Ramachandran plots, not only the numbers in the table.

Reviewer #2 (Remarks to the Author):

The manuscript entitled "Atomic structures of Coxsackievirus A6 and its complex with a neutralizing antibody reveal molecular determinants for vaccine development against hand, foot and mouth disease" by Xu et al reports near atomic resolution cryo-EM structures of the two forms of CVA6 capsids and a CVA6 complex with a neutralizing antibody. The structural analysis of CVA6 is driven by the interest 1) to develop vaccines against CVA6, which causes the significant hand, foot and mouth disease (HFMD) among children, 2) to identify the "footprint" of the neutralizing antibody on the capsid surface and evaluate the likely mechanism of antibody neutralization, and 3) to compare the overall structures as well as the organization of antigenic epitopes of CVA6 with those of other enteroviruses.

The technical aspects of cryo-EM structure determination and associated interpretations are sound. Of course nothing less is expected from the manuscript that comes from the laboratories of star-studded cast of cryo-EM structural biologists (B to Y and Z). The paper is well written with excellent illustrations that are easy to follow.

However, the CVA6 capsid structures reported in the paper are of the empty (84S) and A (altered)-particles (128S). It is puzzling why CVA6 expression did not produce the mature virions (160S). The native (160S) virions have to assemble first and undergo maturation cleavage (VP0 → VP4+VP2) to yield A-particles. Usually the A-particles, which lack VP4 peptides and also exhibit distinct structural characteristics, are formed by the interaction of mature virions (160S) with the cellular receptors. What is the trigger for the promiscuous or spontaneous generation of the A-particles in the case of CVA6 virions? Have the authors considered CVA6 expression in other cells (e.g., Vero cells or HeLa cells). Secondly, it is not clear at what pH the CVA6 particles were expressed/purified and how does it compare with the preparation protocols of other Enteroviruses. Of course getting to the bottom of the absence of mature virions is not the scope of the current paper, but it is potentially interesting.

The authors are encouraged to describe if such a phenomenon of spontaneous generation of A-particles has been observed in other Picornaviruses before and what might be the possible reasons for it as part of the introduction.

It is recommended that the authors use the conventional nomenclature of empty capsids and A-particles, which they are, instead of procapsids (phage nomenclature) and virions.

The details of the Fab structure are minimally described. It is not clear how the amino acid

sequence of Fab (1D5) was obtained. The authors are also encouraged to include a figure showing the tertiary structure of the Fab (a ribbon diagram) as part of figure 5 (similar to the protomer structure as depicted in Fig. 3e).

Others are minor points.

Lines 142-143; strange construction of sentence, consider revising.

Lines 157-158; atomic models were built using the program coot or chimera?

Lines 160-175; when evaluating the structural similarities between different viruses, comparison of only the protomers does not indicate the differences in the quaternary structure. For example similarities of the 2f-channel can be best analyzed by comparing the corresponding 2-fold related protomers (dimer of protomers as a unit) between two capsids and similarly 5-fold related protomer pairs to assess the similarities of the q3f-channel.

Lines: 188-202; not considering the disordered residues, are there significant differences in the organization of N-termini of VP1, VP2 and VP3 (on the inside) of the A-particles from different Picornaviruses.

Line 206; has "nAb" been defined before?

Line 212. Include the saturating number (e.g., 60) of Fab molecules.

Lines 243:247; do the authors imply that the wreath-like Fab coalition is preformed?

Lines 256. What evidence is there that CVA6 A-particles (virions as they are referred to in the paper) bind to cellular receptors. The A-particles of Poliovirus are known to get internalized in a receptor-independent manner (Tuthill et al., JVI 2006, 172-180). Is it possible that Fabs might block the binding of CVA6 A-particles to cells (membranes), instead of blocking their binding to the cellular receptors? Do the results presented in the figure 6 disprove such an argument?

Line 426-427. Remove the words "as described previously" as the references listed correspond to the respective programs.

Line 429. Do the authors mean "building the backbone and side chains from the scratch" when they say "de novo modeling procedure" or some kind of automated model-building procedure was used?

Line 431. Did the authors refine the protomers of CVA6 (in Phenix) or the complete virions (i.e., 60 protomers applying 60-fold strict NCS)? Also why not include the R-factors as part of the Supplementary Table 1.

** Adjust the formatting of the supplementary materials so that headings of the tables appear properly in the same page.

Reviewer #3 (Remarks to the Author):

CVA6 is a known cause of hand foot and mouth disease, for which there are currently no vaccines available and little structural information to guide therapeutics. The isolation and structural characterization of two CVA6 particles are described, along with the identification of 4 loops from interaction with a neutralizing monoclonal antibody. The authors show that the antibody competes with sera raised against the particles indicating that these particles, and especially these 4 loops could be important for vaccine development. The work is well described and will be of interest to the field. The work on genome accessibility, stability and the conformational state of the two particles was well supported. Most of the observations are well-justified.

There are some specific issues that the authors could address to improve the paper:

Results:

The authors conclude that VP4 is missing from the denser particles because it was not seen on SDS-PAGE. There is a possibility that the protein elutes or runs anomalously in the gel, or is lost during purification. Mass spectrometry of the particles is recommended to see if VP4 can be detected prior to concluding that the particle detected is an anomaly.

The authors state that the obvious difference between the two density maps is the presence in virion, but absence in procapsid of RNA. However, they do not show a cross-section of the procapsid reconstruction for the audience to see this. Such a cross section should be included to support this statement.

The authors state that interactions at the 5f with RNA have not been seen in enteroviruses previously (line 201). A low resolution reconstruction of CVA9 has extensively ordered RNA, which could be compared to CVA6 to see if the RNA density is in similar positions. A rough comparison of Fig 2A in Shakeel et al. 2013 and Fig 4a in the current manuscript indicates that this might be the case. Both Ljungan virus & HPeV1 have been reported to also have such ordered RNA, not just HPeV3 and human rhinovirus.

It is not clear to me what is the relevance of ref42 in line 242. Please check this is the correct reference.

Discussion

The claims in 318-320 for a unique RNA binding site and release mechanism for the RNA seem rather speculative and should be toned down. Ironically, for several years, the 5f was considered to be the site of genome release for poliovirus and rhinovirus. Newer data have led to a revision of those ideas.

Methods

The PEG concentration and molecular weight used should be stated (line 343) in order for the work to be reproducible.

Please include a reference to the 2- $\Delta\Delta C$ method (line 392).

Please indicate the source of the Fab used, and the accession number for its sequence.

Supplementary

Supplementary Figure 1, "We will thereafter refer to the the particles" Please correct the typo.

Supplementary Table 2. The authors state that CVA6 procapsid contains VP0 VP1 and VP3, yet the table refers to the capsid proteins VP1 and VP4 for the procapsid. I understand that this makes the comparison of the missing amino acids by residue number easier to compare, but it is inconsistent with some of the main text.

Supplementary Table 3 lower. For completeness, the r.m.s.d. for the surface loops between CVA6 procapsid and VCA6 virion should be given.

The genbank accession number for the CVA6 isolate should also be given in the legend to Supplementary Fig 5.

Response to Reviewer Comments on the manuscript

Manuscript ID: NCOMMS-17-01134A

Summary of responses:

We thank the three reviewers for recognizing the merit of our work and for suggesting ways to improve our paper. As you will see in the following itemized responses, we have addressed all the concerns fully and revised our paper accordingly. We have performed an SDS-PAGE experiment (new Supplementary Figure 1) demonstrating that neither procapsid nor A-particle contains VP4, and added Fig. 4b and Supplementary Figures 4, 5, 9. Throughout the paper, we now refer to the infectious particle to “A-particle” instead of “virion” per suggestion of Reviewer #2. To facilitate the navigation of this document, we have copied the reviewers’ comments verbatim in **blue** and typed our responses in **black**.

Reviewer #1

General opinion and comments:

Reviewer: The manuscript “Atomic structures of Coxsackievirus A6 and its complex with a neutralizing antibody reveal molecular determinants for vaccine development against hand, foot and mouth disease” by Xu and co-authors reports the study of Coxsackie A6 virus (CVA6) using combined approaches: biochemistry, cryo-EM structural analysis, and immunization experiments. This virus is a member of the human Enterovirus species A. The authors report three structures: the CVA6 procapsid, mature CVA6 virion, and the virion-ID5 immune-complex at nearly atomic resolution ($\sim 3.5 \text{ \AA}$). Comparison of these structures demonstrates that loops (BC, DE, EF, and HI) in capsid protein VP1 of CVA6 are involved into interactions with the host cells. This was confirmed by results of biochemical and immune experiments when binding of Fabs induces virus neutralization. The outcome implies that procapsids and heat-treated virions can be used as vaccine candidates to fight HFMD. Structures delivered detailed information on interaction interface between a virus and antibody that is likely suggest means for effective rapid vaccine design.

Images of viral particles were collected on FEI Tecnai F30 cryo microscope equipped with Falcon II camera using EPU automated data collection system. The authors have used Relion software and $\sim 10,000$ images for the procapsid, $\sim 7,000$ for the virions, and $\sim 12,000$ for the virion/Fab complex. The authors have obtained three structures at resolutions allowing to identify residues forming polypeptide chains. Comparison of sequences of capsid proteins from CVA6, EV71, CB3, and EVD68 viruses, their secondary elements, and available atomic structures reveals that all structures are very similar. That is not surprising at such high conservation in polypeptide chain where only specific residues in the VP1 proteins responsible for the neutralizing antigenic epitopes are slightly different which may explain specificity of the viruses. The research presented indicates very nicely how a combination of biochemistry, structural analysis and immunization experiments could lead to the practical applications in medicine.

Specific comments (numbers in reviewer section indicate lines in the original manuscript):

Comment 1: Abstract, line 29. Absence and presence of “RNA interactions”: It is unclear what the authors mean. If they mean interactions with capsid proteins it should be indicated.

Response: “RNA interactions” is now changed to “capsid-RNA interactions” (Page 2, line 25).

Comment 2: Page 4, lines 78-79. I would recommend to remove this sentence, since the authors had a chance to

resolve this “mystery”: they had a sample where they were able to find both empty and filled with genome particles of CVA6 (supl figure 2b). The authors had a wonderful opportunity to find differences between these particles but that was not done. Therefore the sentence does not make sense. Why it was not done it is another question.

Response: It appears that we did not make ourselves clear, causing the misunderstanding. The sentence meant to indicate the motivation behind the results to be presented in this paper. We have revised the sentence to the following (Page 4, lines 65-67): “Though empty and full CVA6 particles have been isolated, there has not been systematic characterization for any of these particles to date and whether they are similar to or different from the above three types of picornavirus particles remains unclear.”

Comment 3: Page 4, lines 81-82. The loops are representing a small percentage of the entire set of sequences of capsid proteins, so it is hardly expected that viruses will be much different in structures and mode of infection. However, specificity to the host cells should be reflected in differences of a few amino acids which are shown by sequence alignments.

Response: Good point. The sentence is now changed to “Though CVA6 shares high (~67%) amino-acid identity with EV71 and CVA16, CVA6 recognizes a different cellular receptor, suggesting CVA6 may differ in its mode of infection, or perhaps even in its local structures involved in infection.” (Page 4, lines 67-70)

Comment 4: Page 5, lines 100 and 104. It seems that the authors have indicated that protein VP4 was not present in the capsids. Explanation should be given in this part related to the both procapsids and mature capsids. Or at least the authors have to prove that VP4 was in procapsids.

Response: Yes, VP4 is absent from both particles. First, in procapsid, VP0 has not been cleaved to yield VP4 as yet. Second, VP4 is already released from our A-particle. We have performed an SDS-PAGE to demonstrate this and the new data is shown in the updated Supplementary Fig.1d.

Comment 5: Page 6, line 124. It would be helpful for a reader if the authors will explain why heating inactivates the virus, otherwise it is confusing.

Response: Heating causes RNA genome to be released, leading to non-infectious empty particle. This is now explained in Page 17, lines 341-342.

Comment 6: Page 7, line 134. “low-dose cryo EM methods” → please rephrase. In EM we do not have different “low-dose methods”. A method is the only one that we use, but we have slightly different technical implementations so it should be the “low-dose imaging” or the “low-dose technique”.

Response: As suggested, “low-dose cryo EM methods” is now changed to “low-dose technique” (Page 6, line 118).

Comment 7: Page 7, line 140. It seems that authors have one average diameter of the capsid but distances between proteins located on 2, 3, and 5-fold axes are vary. It is still not clear between what and what measurements were done.

Response: We now have revised this sentence as follows and added Supplementary Fig.4 to clarify: “Their sizes are the same too, with diameters being 290, 310, 330 Å, along their 2-, 3- and 5-fold axes, respectively (Supplementary Fig.4).” (Page 7, lines 124-125)

Comment 8: Page 7, line 145. I am not sure that RMSD values are very informative for assessment of differences between EM maps. How the authors can prove that these variations above deviations in their measurements? This is an assessment of variations in the pseudo atomic models but not in EM maps. If another person will do the

flexible fit in the same map, what would the RMSD be between two fittings? It is again related to the “golden rule”.

Response: To our knowledge, RMSD (or r.m.s.d) can be used to quantify the difference between two atomic models when they are aligned in most matched interrelated orientation. We agree that RMSD should be generated between atomic models instead of EM maps and a single RMSD value could sometimes be misleading. Actually, a superposition of the two atomic models displayed in Supplementary Fig. 6c demonstrates their excellent match (hence high level of structural similarity) and we revised the text accordingly as follows (Page 7, lines 131-133): “The structural similarity between the atomic models of the two protomers was deemed quite high as judged by the superposition of the two atomic models (Supplementary Fig. 6c) and confirmed as estimated by an r.m.s.d. value of 0.4 Å between the two models.”

Comment 9: Page 7, line 147. The authors like very much the expression “Of note”. I am not sure that it is informative and nearly in all 5-6 sentences within this MS it can be removed without changing the essence of sentences.

Response: We deleted or changed “Of note” in 5 places (Page 3, line 40; Page 7, line 133; Page 9, line 175; Page 12, line 239; Page 14, line 276).

Comment 10: Page 7, line 147. It is unclear to me what the authors implying by saying that VP4 was not seen due to its flexibility. It may be so, but do they have any idea where it can be located? Where to look for it? Did they prove that it was in procapsids?

Response: We meant to indicate that the N-terminal segment of procapsid VP0 (corresponding to the sequence of VP4) is flexible. We have revised the text as follows to prevent confusion (Page 7, lines 133-136). “Though the procapsid was shown by SDS-PAGE to contain VP0 (containing the sequences of VP2 and VP4), only the VP0 sequence corresponding to VP2 can be modeled (Fig. 2d and Supplementary Movie 1), indicating the one corresponding to VP4 is flexible as that in previously reported EV71 procapsid.”

Comment 11: Page 7, line 151-152, It would be good if the authors will rephrase the sentence. It sounds a bit confusing: “... several equivalent or similar regions devoid of residues because corresponding regions of density in the cryoEM maps were smeared...”. I think a situation was reversed: the authors were not able to allocated position of the residues of the loops (that are present in real proteins) due to their flexibility and low local resolution in maps.

Response: The sentence is revised as follows (Page 7, lines 136-139): “Aside from this flexible region, both models contain several additional coincidental flexible segments (Supplementary Fig. 6 and Supplementary Table 2). Among these segments, 7 are identical and the others vary from 2 to 10 amino acid residues (Supplementary Fig. 6a-b and Supplementary Table 2).”

Comment 12: The authors are talking a lot of flexibility of certain areas in capsid proteins, but why not to show the local resolution of these areas? Relion can do that without any extra efforts. That has to be shown on several occasions where the authors are talking of the specific interactions between protein/protein and RNA/protein.

Response: This is done as requested and the results are shown in the new Supplementary Fig.3. The flexible segments described in the paper (lines 139-141) and regions associated with RNA-capsid interactions clearly exhibit lower resolution. The N and C-terminal segments are not visible in these resolution maps, suggesting that they are much more flexible than the other flexible regions mentioned.

Comment 13: Page 7, line 154, It is not a good expression: “... both particles contain a segment of disorder in the

AB loop ...” Should be rephrased.

Response: The revised sentence now reads: “The AB loop of VP2 in the A-particle contains a flexible segment of 10 aa, but 20 aa in the procapsid” (Page 7, lines 140-141).

Comment 14: Page 8, line 160, Sentence “...it has a strikingly similar capsid structure..” why the authors think it is “strikingly” if the protein sequences are nearly identical except loops and very ends of C- and T- termini? See the supl. figure 5.

Response: The word “strikingly” is now deleted (Page 8, line 145).

Comment 15: Page 8, line 173, Not sure that this is the correct expression: “..show loss of ordered residues...” It seems that the authors again were not able to do the fitting and to make an atomic model, since the EM densities were not well defined... but residues were not “lost”.

Response: The sentence is revised as follows (Page 8, lines 158-159): “All three capsid proteins in the CVA6 A-particle show regions of poor densities near the q3f-channel, such as the GH loop...”.

Comment 16: Page 9, line 181, RMSD values do not say anything bad or good. It is not much information here. Again just an indication of flexibility.

Response: Please see also our response to comment 8 above. RMSD values here is to further confirm the difference between A-particle of CVA6 and other enteroviruses which had been shown in Fig. 3e. For clarify the comparison, we also added the RMSD value between CVA6 A-particle and procapsid. Now the sentence reads “The r.m.s.d. values in the C α positions for matched residues in these loops are 1.8 Å, 1.9 Å and 2.8 Å for comparison of the CVA16, EV71 A-particles and EV71 mature virus, respectively, while the value is just 0.3 Å in comparison to CVA6 procapsid (Supplementary Table 3);” (Page 9, lines 165-167).

Comment 17: Pages 9-10, lines 188-202. The section related to Capsid /RNA interaction is very confusing. The authors show maps at extremely low threshold and a very thick slab, so one can put anything into these bulky areas of density. It is puzzling how the individual residues of VP2 that interact with genome were identified? Especially when the authors write correctly that there is a mismatch between the capsid proteins and RNA within the capsid: capsid proteins have symmetrical arrangement while genome is asymmetrical. Demonstration of the local resolution here will be helpful since the authors claim exact amino acids of the chain that are in the contact with RNA, while in the pictures a resolution seems to be hardly 10 Å.

Response: The confusion stems from our old figure showing a low resolution density map superposed with the atomic model. We now replace this figure with our high resolution capsid density map where individual amino acid residues can be registered with the atomic model (new Fig.4). We also include a local resolution map (new Fig.4b) as requested. The related text has been revised accordingly (see lines 172-187).

Comment 18: Page 10, line 206. Sizes of the Fab used should be given both as M.m, and length in AA. That will help to asses quickly changes in sizes and its visibility on cryo images.

Response: MW and aa length of Fab-1D5 were added in our revised manuscript, and it now reads “(MW = ~50kDa, 108aa and 119aa corresponding to light and heavy chains of the variable domain)” (see Page 18, line 383). We didn't obtain the sequence of constant domain in Fab-1D5.

Comment 19: Page 10, line 211. The authors write that densities corresponding to the Fab are well defined. Possibly yes..., however, it does not look very convincing in the supl figure 2c. The authors did not provide any

information on how the classes were obtained, how many images are in each class. Although it is natural to assume that the classes were obtained in Relion, then the classes would comprise up to several hundreds of images; so it would be risky to say that densities that protrude from the capsid in projections verify full occupancy of binding sites with the Fabs. Moreover, the densities seen as protrusion around the capsids are not so strong as densities corresponding to the capsids. So there are some doubts that all binding sites on the surface of the capsids are occupied by Fabs.

Response: We agree with the reviewer that the virus-fab complex don't exhibit claiming full occupancy for the Fabs. The sentence is revised and now reads (Page 10, lines 195-196) "2D classification of such images show well-defined Fab density features that extend ~60 Å above the CVA6 surface (Supplementary Fig.2d)."

Comment 20: Page 10, line 214. It is not very usual expression "an estimated resolution limit.". Typically we are talking of a resolution of a structure but not of the resolution limit in the structure. The resolution can be limited by different factors.

Response: The expression of "an estimated resolution limit." is changed to: "an estimated resolution of 3.8 Å" (Page 10, line 197-198).

Comment 21: Page 10, lines 217-220. The authors did not provide definitions of "constant", "variable", rigid, and flexible domains. It seems that sometimes they are mixed up themselves with flexibility and variability, but variable domains have "approximately" the same densities as the capsid. The authors have to make clear what they want to say.

Response: The "variable domain" and "constant domain" are conventional terms in antibody field, corresponding to variable and relatively conserved in aa sequence of evolutionary antibodies. We now have revised this sentence to clarify "The variable domains (directly contacting the capsid epitope) of the bound Fab molecules exhibit density approximately as strong as that of the capsid shell, which supports the notion that the Fab sites on the capsid are nearly fully occupied. Also, as expected owing to the flexible connection between constant and variable domains in Fabs, the constant domains exhibit weaker density compared to the capsid shell and the Fab variable domains (Fig. 5b and Supplementary Fig.9a)." (Page 10, lines 201-205). We also added a new supplementary Fig.9 that exhibits the constant and variable domains of the Fab molecule.

Comment 22: Page 10, line 219. Please rephrase: "...the Fab sites on the capsid are saturated with high binding affinity...". Should it be? : "...To assess saturation of higher-affinity binding sites ,, experiment were performed..." or whatever was done by authors to assess the occupancy of Fabs. May be the authors had something different in their minds.

Response: The near atomic resolution (3.8Å) of cryoEM map of complex and the strong density of variable domain of Fab (nearly as strong as capsid) can well demonstrate the full occupancy of Fabs. But we indeed didn't perform any other experiment to assess the occupancy of Fabs, so we have tone down the text to prevent confusion, please see our response to **Comment 21** above.

Comment 23: Page 11, line 226. "... Each Fab contacts a footprint.." Should it be "...identification of a Fab interaction footprint site on a virus surface was done by..."

Response: The revised sentence now reads "Identification of a Fab interaction footprint site on the virus surface was performed by the PISA program, and the interaction interface of each Fab covers an area of 675 Å²" (Page 11, lines 209-210).

Comment 24: Page 11, line 237. The authors are writing about H-bonds between light chains of Fabs. Do they really see H-bonds? It would be good to see these areas of maps and indications of local resolution.

Response: The ResMap result showed that the local resolution in this region is $\sim 3.6 \text{ \AA}$ (new supplementary figure 3), which allows us to accurately build the side chains of amino acids D1 and Y66. Identification of H-bonds in our atomic models was done by the “FindHBond” tool in Chimera. H-bond identification in atomic models derived from near atomic resolution structures has many precedencies, as in *Wang et al., PNAS 2016* and *Shanker et al., PNAS. 2016*.

Comment 25: Page 12, line 244. Here is very unusual sentence: “...such wreath-like Fab coalition covers a huge footprint and hence the binding affinity is also enhanced by approximately 5-fold”. It should be rephrased.

Response: The sentence now reads “... such wreath-like five-Fabs coalition covers a huge footprint (an area of 3375 \AA^2) on the A-particle capsid surface, which is consistent with the high binding affinity of 1D5.” (Page 11, lines 227-229).

Comment 26: Page 12, line 246, The authors write that “a well saturated and rigid immune-complex, which is proved to be advantageous for the atomic resolution solution” ... The sentence has to be rephrased. Even more, it contradicts to the sentence on the page 10 (line 220) where constant domains have weak densities and therefore were not resolved.

Response: Sorry for the confusion. The weak densities of constant domain of Fab was caused by flexible connection between constant and variable domain. We have revised the sentence as follows to prevent confusion: “This might explain why the CVA6 A-particle-1D5 is a rather stable immune-complex, which enable its structure to be determined at high resolution.” (Page 11-12, lines 229-230)

Comment 27: Page 12, lines 252-253. It is not very clear if the full occupancy by Fabs on capsids is required to prevent binding of viral particles to the host cells. Possibly not.

Response: We certainly agree that full occupancy may not be required to prevent binding of the viral particles to the host cells. However, we are currently only able to prove that 1D5 can significantly reduce the number of virus particles bound to host cells based on the RT-PCR experiment.

Comment 28: Page 12, line 264. RMSD numbers do not provide any Info on the similarity between the structures, it is a level of similarity between atomic models and possibly may reflect only the accuracy of the fitting within a given map at a given resolution. The lower resolution -> more variations in the fitting.

Response: See also our response to comment 8 above. The atomic models were built de novo from the 3.8 \AA resolution map and the accuracy of these models is at the level of individual amino acids. We now qualified the accuracy of structural arrangement by this resolution (see Page 12, lines 242-245), as follows: “At least up to the resolution (3.8 \AA) of our cryoEM map, Fab-1D5 binding induces no significant structural rearrangement of the CVA6 capsid as revealed by their superposition of the protomers in the CVA6-A-particle-1D5 complex and in A-particle and also reflected by the r.m.s.d. value of 0.4 \AA between the $C\alpha$ atoms of the two structures (supplementary Fig. 9c and Supplementary Table 3).”

Comment 29: Page 14, lines 293-301. The authors are discussing the variation in positions of loops and claiming that there are significant (prominent?) differences. So again there is some contradiction: structures are nearly the same that have very low RMSD, nonetheless there are significant differences in loop locations. How are big the differences? How they were assessed. The amplitude of variations should be shown in figures. The authors are

saying that the loops are involved in the interactions with receptors, but on the other side their sequences make the loops specific at recognition of receptors of the host cells. What else is changing in the loops in different viruses apart of their positions?

Response: The original lines 293-301 describe how the loops differ across CVA6, EV71 and CVA16 and other than between different particles of CVA6. Thus, there is no contradiction. The loops in question are well resolved in our cryoEM density map and their models are accurate to the resolution of the map (3.3Å).

Comment 30: Page 15, lines 314-320. Release of RNA. The hypothesis is very confusing and not properly explained. The authors write that RNA has strong interactions with 5-fold vertices of the capsid; and therefore release of RNA can take place via 5-fold vertices. It does not sound as a logical hypothesis. The Fabs may block or arrest release of RNA from capsids as it was demonstrated by the biochemical experiments, In these type of viruses with T=3 symmetry Fabs have only this option since the binding sites are located around 5-fold axes. However how genome will be released remains unclear.

Response: We fully agree with the reviewer here about our hypothesis and have removed the sentence “Thus, the CVA6 virion might release its genome through a 5f-channel, and, if true, would represent a unique infection mechanism for an enterovirus” (Page 15, line 297).

Comment 31: Page 20, line 431 “Atomic positions of residues were built and adjusted inside Coot”. The sentence has to be rephrased... I hope that is was not done “inside Coot” but using .a software package Coot. I did not find the reference for Coot while Chimera has the reference.

Response: The sentence is changed to: “Atomic positions of residues were built and modified using the Coot software package”. The reference for Coot (ref 58) was in the revised manuscript (line 409). (Page20, lines 412-413)

Comment 32: Figure 2. Panels a,b, and c are too small. Possibly it makes sense to show a part of the capsid and make a link with the asymmetrical subunit. Panel d should be moved to Figure 3 and combined with figure3 e -: all three models that have to be fitted into this asymmetrical unit and amplitudes of loop displacements have to be indicated.

Response: Done as suggested. Note, panel d is the asymmetric unit so is now linked to the capsid in panel c (as suggested), and is not moved to Fig.3.

Comment 33: Figure 3. Should be partly combined with Figure 2. See comments above.

Response: Please see the response to **Comment 32**.

Comment 34: Figure 4. A threshold used to show a surface of the 3D map is extremely low and does not indicate a resolution at which some amino acids can be resolved. The authors have claimed resolution ~ 3.5 Å, beta strands have to be clearly resolved, but in this figure it is definitely not the case. The authors indicating specific residues those are supposed to interact with RNA. But in this map they are not seen. It seems that these AA were taken from other structures. The AA are not shown in the “ball-and stick-mode” but represented as ball and sticks. The authors have mixed up letters used for designation of panels.

Response: Thank you for pointing this out. We now replace this figure with our high resolution capsid density map where individual amino acid residues can be registered with the atomic model (new Fig.4), in the new figure, capsid-RNA interaction can be observed obviously and the key aa involved in interaction are now shown as spheres. Please see also the response to **comment 17** above.

Comment 35: Figure 5. “Iso-contoured views” -> should be views of an isosurface. At which threshold they are shown? “...the constant domain has disappeared...” ? It has not, it was not resolved. The explanation of Fab organization has to be given in the main text. The domains have to be labelled in the figure. What is a difference between “close-up views” and “magnified in views”?

Response: “Iso-contoured views” is now changed to “isosurface views”, “magnified in views” is now changed to “expanded views”. (Page 37, line 631, 638)

Comment 36: Suppl Figure 2. Panel d- more information should be provided on classes and how they have been obtained. Images look like they are harshly coarsened projections of a 3D reconstruction of the virion/Fab complex: they have very good mirror or close to 2-fold rotational symmetries. It is a bit strange. However the densities of Fabs do not say much about the Fab occupancy.

Response: We have added more information for Supplementary Fig. 2d as “Representative 2D class averages of CVA6 A-particle-1D5 complex particle images. A total of 34 well-aligned classes (12067 particles) were selected for subsequent 3D classification, and the first 10 mostly populated classes (6386 particles) are shown in (d). Fab densities are clearly visible surrounding the outside of the virus capsid shell. White represents protein or genome density in (d).”

Comment 37: Suppl Figure 3. This is the most puzzling figure. Panels a, b, and c represent the Fourier Shell correlation functions and they are strange: they are cropped from the bottom. The curve behavior indicates that FSC values became negative after crossing X axis which is very strange. That can take place if the maps in comparison have the opposite contrast at high frequencies? It means that the authors have applied a bit strange filter on maps, or something else went wrong during masking of 3D maps. The FSC should be shown in full without any truncations. Panels b, d, and f -> the local resolution maps are too small, they are tightly masked and show only surfaces at the same local resolution. What was the threshold used to show isosurfaces? The authors have to show cut-away central views of reconstructions and possibly only one quarter, so magnification will be sufficient to see variation in resolution.

Response: We did not crop the FSC data, which were output automatically from Relion 1.4 and plotted by software Prism directly. Recognizing this reviewer’s concern, we installed the most updated Relion software (v2.0) and subjected the two unfiltered half maps obtained previously to Relion 2.0’s post-processing and local resolution tools. The new (untruncated) FSC curves are shown in new Supplementary Fig.3 and they confirm our previous plots. Local resolution maps were displayed automatically without user input for isosurface threshold. These maps are now enlarged and their open views (one asymmetric unit) are also provided (new Supplementary Figure 3). See also response to Comment 12 above.

Comment 38: It would helpful be to see (in supplementary materials) Ramachandran plots, not only the numbers in the table.

Response: Ramachandran plots results are now added in Supplementary information (see Supplementary Fig. 5).

Reviewer #2

General opinion and comments:

Reviewer: The manuscript entitled “Atomic structures of Cocksackievirus A6 and its complex with a neutralizing

antibody reveal molecular determinants for vaccine development against hand, foot and mouth disease” by Xu et al reports near atomic resolution cryo-EM structures of the two forms of CVA6 capsids and a CVA6 complex with a neutralizing antibody. The structural analysis of CVA6 is driven by the interest 1) to develop vaccines against CVA6, which causes the significant hand, foot and mouth disease (HFMD) among children, 2) to identify the “footprint” of the neutralizing antibody on the capsid surface and evaluate the likely mechanism of antibody neutralization, and 3) to compare the overall structures as well as the organization of antigenic epitopes of CVA6 with those of other enteroviruses.

The technical aspects of cryo-EM structure determination and associated interpretations are sound. Of course nothing less is expected from the manuscript that comes from the laboratories of star-studded cast of cryo-EM structural biologists (B to Y and Z). The paper is well written with excellent illustrations that are easy to follow.

Comment 1: However, the CVA6 capsid structures reported in the paper are of the empty (84S) and A (altered)-particles (128S). It is puzzling why CVA6 expression did not produce the mature virions (160S). The native (160S) virions have to assemble first and undergo maturation cleavage (VP0 → VP4+VP2) to yield A-particles. Usually the A-particles, which lack VP4 peptides and also exhibit distinct structural characteristics, are formed by the interaction of mature virions (160S) with the cellular receptors. What is the trigger for the promiscuous or spontaneous generation of the A-particles in the case of CVA6 virions? Have the authors considered CVA6 expression in other cells (e.g., Vero cells or HeLa cells). Secondly, it is not clear at what pH the CVA6 particles were expressed/purified and how does it compare with the preparation protocols of other Enteroviruses. Of course getting to the bottom of the absence of mature virions is not the scope of the current paper, but it is potentially interesting.

Response: We are also puzzled by the observation that CVA6 did not produce mature virions. Like the situation for EV71 and CVA16 (*Lyu et al. J Virol. 2013* and *Fan et al. J Virol. 2017*), in this study CVA6 was grown in human rhabdomyosarcoma (RD) cells, purified following the same protocols and stored in PBS (pH 7.4) buffer.

Comment 2: The authors are encouraged to describe if such a phenomenon of spontaneous generation of A-particles has been observed in other Picornaviruses before and what might be the possible reasons for it as part of the introduction.

Response: The atomic structure of the A-particle of CVA16 was reported (see *Ren et al. Nat Commun. 2013*). However, it remains unclear why these A-particle only mysteriously emerged in only one batch out of many sample preparations. The above content was mentioned in the discussion section but not in the introduction (Page 13, lines 262-264) to keep the logical flow of the texts in the introduction.

Comment 3: It is recommended that the authors use the conventional nomenclature of empty capsids and A-particles, which they are, instead of procapsids (phage nomenclature) and virions.

Response: As suggested, the term “virion” is changed to “A-particle” throughout the manuscript. We keep the nomenclature “procapsid” to distinguishing from empty particle that transformed from mature virus. The nomenclature of “procapsid” is also used in EV71 (*Shigler et al. PLoS Pathog. 2013* and *Cifuentes et al. J Virol. 2013*).

Comment 4: The details of the Fab structure are minimally described. It is not clear how the amino acid sequence of Fab (1D5) was obtained. The authors are also encouraged to include a figure showing the tertiary structure of the Fab (a ribbon diagram) as part of figure 5 (similar to the protomer structure as depicted in Fig. 3e).

Response: This is done as requested and this part of the description now reads “Anti-CVA6 monoclonal antibody (1D5) was screened and produced using the methods previously described. The RNA was isolated from 1D5 hybridoma cells and converted to cDNA by reverse transcription. Then, the variable regions of the heavy chain and light chain of 1D5 were amplified by PCR for sequence determination.” (Page 22 in revised supplementary information). In addition, a figure showing the tertiary structure of the Fab was added into Supplementary Fig.9.

Specific comments (numbers in reviewer section indicate lines in the original manuscript):

Comment 5: Lines 142-143; strange construction of sentence, consider revising.

Response: The sentence is now changed to “The obvious difference between two density maps is the density of genomic RNA, which is present in A-particles but absent in procapsids (Supplementary Fig.4)” (Page 7, Line 127-128).

Comment 6: Lines 157-158; atomic models were built using the program coot or chimera?

Response: Atomic models of protomers were built with Coot and refined with Phenix (page 20, line 408), then we used program Chimera to generate entire capsids. The sentence is now changed to: “Atomic models of entire capsids (Fig. 2c), except for the disordered regions identified above, were generated using the program Chimera”. (Page 8, lines 143-144).

Comment 7: Lines 160-175; when evaluating the structural similarities between different viruses, comparison of only the protomers does not indicate the differences in the quaternary structure. For example similarities of the 2f-channel can be best analyzed by comparing the corresponding 2-fold related protomers (dimer of protomers as a unit) between two capsids and similarly 5-fold related protomer pairs to assess the similarities of the q3f-channel.

Response: The selected 4 protomers quaternary structure (Fig. 3a-d) contain full information of 2f- and q3f-channels. Such comparison was also performed in previous studies on EV71 and CVA16 (*Xiangxi Wang et al. Nat. Struct. Mol. Biol. 2012; Jingshan Ren et al. Nat. Commun. 2013*).

Comment 8: Lines: 188-202; not considering the disordered residues, are there significant differences in the organization of N-termini of VP1, VP2 and VP3 (on the inside) of the A-particles from different Picornaviruses.

Response: There are no significant differences in the organization of N-termini of VP1, VP2 and VP3 (on the inside) of the A-particles from different Picornaviruses. The sequence alignment indicates that N-termini of CVA6 share several conservative regions with EV71 and CVA16 (see supplementary Fig 7). In addition, A-particles of both EV71 and CA16 have similar disordered regions in N-termini of VP1 and VP2 as CVA6 does (*Lyu et al. J Virol. 2013; Ren et al. Nat. Commun. 2013*). Beside the above similarities, N-termini (62-72 aa) of VP1 in CVA16 A-particle crystal structure are found traversing the capsid shell to the outside, which is not yet found in EV71 A-particle or CVA6 A-particle.

Comment 9: Line 206; has “nAb” been defined before?

Response: We defined the “nAb” as “neutralizing antibody (nAb)” in our revised manuscript (Page 10, line 190).

Comment 10: Line 212. Include the saturating number (e.g., 60) of Fab molecules.

Response: The 3.8 Å resolution map shows that a total of 60 Fab molecules can bind to each capsid, which has been mentioned in the following description. Now the sentence reads “2D classification of such images show well-defined Fab density features that extend ~60 Å above the CVA6 surface (Supplementary Fig. 2d).” (Page 10, lines 195-196)

Comment 11: Lines 243:247; do the authors imply that the wreath-like Fab coalition is preformed?

Response: Actually not. Such wreath-like five-Fabs coalition is believed to be formed by inter-Fab interaction (H-bound) when 5 Fab molecules binding to one 5-fold vertex.

Comment 12: Lines 256. What evidence is there that CVA6 A-particles (virions as they are referred to in the paper) bind to cellular receptors. The A-particles of Poliovirus are known to get internalized in a receptor-independent manner (Tuthill et al., JVI 2006, 172-180). Is it possible that Fabs might block the binding of CVA6 A-particles to cells (membranes), instead of blocking their binding to the cellular receptors? Do the results presented in the figure 6 disprove such an argument?

Response: Good point. The results in Fig.6 only demonstrate that antibodies block contact between CVA6 and cell surface. Thus, we cannot distinguish between receptor mediated and receptor independent mechanisms of cell entry. We have revised the text accordingly (Page 12, lines 233-241).

Comment 13: Line 426-427. Remove the words “as described previously” as the references listed correspond to the respective programs.

Response: “as described previously” is deleted. (Page 20, lines 408-409).

Comment 14: Line 429. Do the authors mean “building the backbone and side chains from the scratch” when they say “de novo modeling procedure” or some kind of automated model-building procedure was used?

Response: Yes, here, the de novo modeling refers to the procedure of building the backbone and side chains of amino acids in the atomic model from the scratch.

Comment 15: Line 431. Did the authors refine the protomers of CVA6 (in Phenix) or the complete virions (i.e., 60 protomers applying 60-fold strict NCS)? Also why not include the R-factors as part of the Supplementary Table 1.

Response: Here, we used the tool of real_space_refine in Phenix to refine the protomers of both CVA6 A-particle and its immune-complex without applying 60-fold strict NCS. This method does not report R-factors in the final results. We also noticed that R-factors were not reported in many other colleagues’ work, such as *Corey F.Hryc et al., PNAS 2017*, *Zhao Wang et al. Nature communications 2014*, *Karl V. Gorzelnik et al. PNAS 2016*, *Xiangxi Wang et al. PNAS 2016*, etc.

Comment 16: Adjust the formatting of the supplementary materials so that headings of the tables appear properly in the same page.

Response: Done.

Reviewer #3

General opinion and comments:

Reviewer: CVA6 is a known case of hand foot and mouth disease, for which there are currently no vaccines available and little structural information to guide therapeutics. The isolation and structural characterization of two CVA6 particles are described, along with the identification of 4 loops from interaction with a neutralizing monoclonal antibody. The authors show that the antibody competes with sera raised against the particles indicating that these particles, and especially these 4 loops could be important for vaccine development. The work is well

described and will be of interest to the field. The work on genome accessibility, stability and the conformational state of the two particles was well supported. Most of the observations are well-justified.

Specific comments (numbers in reviewer section indicate lines in the original manuscript):

Results:

Comment 1: The authors conclude that VP4 is missing from the denser particles because it was not seen on SDS-PAGE. There is a possibility that the protein elutes or runs anomalously in the gel, or is lost during purification. Mass spectrometry of the particles is recommended to see if VP4 can be detected prior to concluding that the particle detected is an anomaly.

Response: VP4 is absent from both particles. First, in procapsid, VP0 has not been cleaved to yield VP4 as yet. Second, VP4 is already released from our A-particle. We have performed an SDS-PAGE to demonstrate this and the new data is shown in the updated Supplementary Figure 1d.

Comment 2: The authors state that the obvious difference between the two density maps is the presence in virion, but absence in empty particle of RNA. However, they do not show a cross-section of the procapsid reconstruction for the audience to see this. Such a cross section should be included to support this statement.

Response: As suggested, we have added the central cross-section views of both CVA6 procapsid and A-particle in Supplementary Fig.4. (Page 7 in supplementary information).

Comment 3: The authors state that interactions at the 5f with RNA have not been seen in enteroviruses previously (line 201). A low resolution reconstruction of CVA9 has extensively ordered RNA, which could be compared to CVA6 to see if the RNA density is in similar positions. A rough comparison of Fig 2A in Shakeel et al. 2013 and Fig 4a in the current manuscript indicates that this might be the case. Both Ljungan virus & HPeV1 have been reported to also have such ordered RNA, not just HPeV3 and human rhinovirus.

Response: In CVA9, the most obvious contacts below the 5-fold vertex involve the N and C termini of VP4 and the N terminus of VP1 (Shakeel et al. J Virol. 2013). However, in contrast, in this study the strong interactions between the N-termini of VP3 and the genomic RNA at the icosahedral 5-fold axes were apparent. In addition, ordered RNA segments in the structures of Ljungan virus and HPeV1 have been added in our revised manuscript, and now reads “Similar interactions between the VP3 N-termini and the genome close to the icosahedral 5-fold axes have been reported in picornavirus Ljungan virus, HPeV1, HPeV3 and human rhinovirus but have never been observed in any of the enterovirus species.” (Page 9, lines 183-185).

Comment 4: It is not clear to me what is the relevance of ref42 in line 242. Please check this is the correct reference.

Response: It is the correct reference. In order to separate the units (\AA^2) and references (42), we added letters “ref” before 42. Now, “ref” is removed (Page 11, line 226).

Discussion:

Comment 5: The claims in 318-320 for a unique RNA binding site and release mechanism for the RNA seem rather speculative and should be toned down. Ironically, for several years, the 5f was considered to be the site of genome release for poliovirus and rhinovirus. Newer data have led to a revision of those ideas.

Response: We fully agree with the reviewer here about our hypothesis. Now the sentence “Thus, the CVA6 virion might release its genome through a 5f-channel, and, if true, would represent a unique infection mechanism for an enterovirus” is deleted (Page 15, Line 297).

Methods:

Comment 6: The PEG concentration and molecular weight used should be stated (line 343) in order for the work to be reproducible.

Response: The sentence is now revised accordingly: “Virus was harvested 3 days post infection, centrifuged to remove cell debris, ultrafiltered. The virus supernatant was mixed with 50% polyethylene glycol 8000 (PEG 8000) and 2 M NaCl-phosphate-buffered saline (PBS) (pH 7.4) to final concentrations of 8% and 300 mM, respectively, and stirred overnight at 4°C. After centrifugation and removing the pellet, the CVA6 virus particles were loaded onto a 15%-45% (w/v) sucrose density gradient and centrifuged at 30,000 rpm for 4.5 h in a Beckman SW41 rotor at 4 °C.” (Page 15-16, lines 317-322).

Comment 7: Please include a reference to the 2- $\Delta\Delta C$ method (line 392).

Response: Done (Page 18, line 372).

Comment 8: Please indicate the source of the Fab used, and the accession number for its sequence.

Response: The procedure of producing and screening 1D5 was added in the section of supplementary methods. The 1D5 does not have the accession number and is obtained from our laboratory. Now this part of the description reads “Anti-CVA6 monoclonal antibody (1D5) was screened and produced using the methods previously described. The RNA was isolated from 1D5 hybridoma cells and converted to cDNA by reverse transcription. Then, the variable regions of the heavy chain and light chain of 1D5 were amplified by PCR for sequence determination.” (Page 22, lines 187-190 in supplementary information).

Supplementary:

Comment 9: Supplementary Figure 1, “We will thereafter refer to the the particles” Please correct the typo.

Response: Done (Page 3, Line 38).

Comment 10: Supplementary Table 2. The authors state that CVA6 procapsid contains VP0 VP1 and VP3, yet the table refers to the capsid proteins VP1 and VP4 for the procapsid. I understand that this make the comparison of the missing amino acids by residue number easier to compare, but it is inconsistent with some of the main text.

Response: As suggested, the supplementary Table 2 is revised (Page 17 in Supplementary Information).

Comment 11: Supplementary Table 3 lower. For completeness, the r.m.s.d. for the surface loops between CVA6 procapsid and VCA6 virion should be given.

Response: The r.m.s.d. for the surface loops is now given in the Supplementary Table 3 (Supplementary page 18).

Comment 12: The genbank accession number for the CVA6 isolate should also be given in the legend to Supplementary Fig 5.

Response: The genbank accession number for the CVA6 is now given in the legend to Supplementary Fig.7 (Supplementary page 11, line 94).

REVIEWERS' COMMENTS:

Reviewer #1 (Remarks to the Author):

The manuscript was significantly improved. The text and figures were modified according to requests by reviewers. In general I am satisfied with majority of changes done by the authors.

There are still some linguistic problems such as “..VP0 (containing the sequences of VP2 and VP4), only the VP0 sequence corresponding to VP2 can be modeled...”

Protein VP0 comprises two major domains corresponding to VP2 and VP4. Apparently VP4 is cleaved? would it be a correct idea? The authors have to explain what do they meant as “modeling”. Do they mean prediction of secondary elements, or tracing the chain? There are some other places where similar difficulties arise at the MS reading,

An additional proof reading should be done after the corrections will be finished.

Reviewer #2 (Remarks to the Author):

None

=====

Reviewer #1

General opinion and comments:

Reviewer: The manuscript was significantly improved. The text and figures were modified according to requests by reviewers. In general I am satisfied with majority of changes done by the authors.

There are still some linguistic problems such as “..VP0 (containing the sequences of VP2 and VP4), only the VP0 sequence corresponding to VP2 can be modeled...”

Protein VP0 comprises two major domains corresponding to VP2 and VP4. Apparently VP4 is cleaved? would it be a correct idea? The authors have to explain what do they meant as “modeling”. Do they mean prediction of secondary elements, or tracing the chain? There are some other places where similar difficulties arise at the MS reading.

An additional proof reading should be done after the corrections will be finished.

Response: In procapsid, VP0 has not yet been cleaved to yield VP2 and VP4. “Can be modeled” means to build the atomic model (“tracing the chain”) of the protein. We have revised the text as follows to prevent confusion (Page 7, lines 133-137). “Though the procapsid was shown by SDS-PAGE to contain VP0 (containing the sequences of VP2 and VP4), only the VP0 sequence corresponding to VP2 is well resolved in the density map and hence successfully modeled, indicating the one corresponding to VP4 is flexible as that in previously reported EV71 procapsid.”